# Evolution of multicellularity by collective integration of spatial information

Enrico Sandro Colizzi[1]*, Renske MA Vroomans[2], Roeland MH Merks[3]

[1]Mathematical Institute, Leiden University; Origins Center, Leiden, Netherlands; [2]Informatics Institute, University of Amsterdam; Origins Center, Amsterdam, Netherlands; [3]Mathematical Institute, Leiden University; Institute of Biology, Leiden University; Origins Center, Leiden, Netherlands

**Abstract** At the origin of multicellularity, cells may have evolved aggregation in response to predation, for functional specialisation or to allow large-scale integration of environmental cues. These group-level properties emerged from the interactions between cells in a group, and determined the selection pressures experienced by these cells. We investigate the evolution of multicellularity with an evolutionary model where cells search for resources by chemotaxis in a shallow, noisy gradient. Cells can evolve their adhesion to others in a periodically changing environment, where a cell's fitness solely depends on its distance from the gradient source. We show that multicellular aggregates evolve because they perform chemotaxis more efficiently than single cells. Only when the environment changes too frequently, a unicellular state evolves which relies on cell dispersal. Both strategies prevent the invasion of the other through interference competition, creating evolutionary bi-stability. Therefore, collective behaviour can be an emergent selective driver for undifferentiated multicellularity.

## Introduction

The evolution of multicellularity is a major transition in individuality, from autonomously replicating cells to groups of interdependent cells forming a higher-level of organisation (*Buss, 2014*; *Smith and Szathmary, 1995*). It has evolved independently several times across the tree of life (*Grosberg and Strathmann, 2007*; *Parfrey and Lahr, 2013*). Comparative genomics suggests (*Knoll, 2011*), and experimental evolution confirms (*Boraas et al., 1998*; *Ratcliff et al., 2012*) that the increase of cell–cell adhesion drives the early evolution of (undifferentiated) multicellularity. Increased cell adhesion may be temporally limited and/or may be triggered by environmental changes (e.g. in Dictyostelids and Myxobacteria [*Du et al., 2015*; *Kaiser et al., 1979*]). Moreover, multicellular organisation may come about either by aggregation of genetically distinct cells or by incomplete separation after cell division (*King, 2004*; *Du et al., 2015*).

The genetic toolkit and the cellular components that allow for multicellularity - including adhesion proteins - pre-date multicellular species and are found in their unicellular relatives (*Rokas, 2008*; *Prochnik et al., 2010*; *Du et al., 2015*; *Richter et al., 2018*). Aggregates of cells can organise themselves by exploiting these old components in the new multicellular context, allowing them to perform novel functions (or to perform old functions in novel ways) that may confer some competitive advantage over single cells. Greater complexity can later evolve by coordinating the division of tasks between different cell lineages of the same organism (e.g. in the soma-germline division of labour), giving rise to embryonic development. Nevertheless, the properties of early multicellular organisms are defined by self-organised aggregate cell dynamics, and the space of possible multicellular outcomes and emergent functions resulting from such self-organisation seems large – even with limited differential adhesion and signalling between cells. However, the evolution of emergent functions as a consequence of adhesion-mediated self-organisation has received little attention to date.

*For correspondence:
e.s.colizzi@math.leidenuniv.nl

**Competing interests:** The authors declare that no competing interests exist.

**eLife digest** All multicellular organisms, from fungi to humans, started out life as single cell organisms. These cells were able to survive on their own for billions of years before aggregating together to form multicellular groups. Although there are trade-offs for being in a group, such as sharing resources, there are also benefits: in a group, single cells can divide tasks amongst themselves to become more efficient, and can develop sophisticated mechanisms to protect each other from harm. But what compelled single cells to make the first move and aggregate into a group?

One way to answer this question is to study the behaviour of slime moulds. These organisms exist as single cells but form colonies when their resources run low. Researchers have observed that slime mould colonies can navigate their environment much better than single cells alone. This property suggests that the benefits of moving together as a collective could be the driving factor propelling single cells to form groups.

To test this theory, Colizzi et al. developed a computer model to examine how well groups of cells and lone individuals responded to nearby chemical cues. Unlike previous simulations, the model created by Colizzi et al. did not specify that being in a group was necessarily more favourable than existing as a single cell. Instead, it was left for evolution to decide which was the best option in response to the changing environmental conditions of the simulation.

The mathematical model showed that groups of cells were generally better at sensing and moving towards a resource than single cells acting alone. Single cells moved at the same speed as groups, but they often sensed their environment poorly and got disorientated. Only when the environment changed frequently, did cells revert back to the single life. This was because it was no longer beneficial to band together as a group, as the cells were unable to sense the environmental cues fast enough to communicate to each other and coordinate a response.

This work provides insights into what drove the early evolution of complex life and explains why, under certain conditions, single cells evolved to form colonies. Additionally, if this model were to be adopted by cancer biologists, it could help researchers better understand how cancer cells form groups to move and spread around the body.

Mathematical models can define under which conditions multicellularity evolves, in terms of fitness for individual cells vs. the group, or in terms of the resulting spatial and temporal organisation. The formation of early multicellular groups has been studied in the context of the evolution of cooperation: by incorporating game theoretical interactions and transient compartimentalisation (*Garcia et al., 2014*) or the possibility of differential assortment (*Joshi et al., 2017*), it was found that adhering groups of cooperating individuals evolve. Alternatively, reproductive trade-offs can give rise to division of labour (*Solari et al., 2013*) and lead to the formation of a higher-level proto-organism capable of self-regeneration in a structured environment (*Duran-Nebreda et al., 2016*). A plethora of multicellular life-cycles can emerge by simple considerations about the ecology of the uni-cellular ancestor and the fitness benefit that cells acquire by being in groups (*Staps et al., 2019*). Once multicellular clusters are established, the spatial organisation of their composing cells can play an important role in determining group-level reproduction - possibly leading to the evolution of cell-death (*Libby et al., 2014*) or different cell shapes (*Jacobeen et al., 2018*), and to specific modes of fragmentation of the aggregate (*Pichugin et al., 2017*; *Gao et al., 2019*) that increase overall population growth.

In these models, multicellularity is either presupposed or its selective pressure is predetermined by social dynamics, by directly increasing fitness of cells in aggregates or by adverse environmental conditions that enforce strong trade-offs. Here we investigate the origin of this selective pressure, motivated by the idea that multicellular groups emerge as a byproduct of cell self-organisation and cell-environment interactions, and subsequently alter the evolution of their composing cells. We expect that a selective pressure to aggregate can arise from the emergent functions of the multicellular group, without requiring explicit selective advantages and disadvantages for cells in a group. We therefore present a computational model of an evolving population of cells where fitness is

based solely on how adequately a cell responds to a spatially and temporally heterogeneous environment, regardless of whether they belong to an aggregate.

In this study, we draw inspiration from collective movement of groups of cells, such as the aggregate phase of the slime mould *Dictyostelium discoideum* (*Schaap, 2011*), other simple multicellular organisms (*Kaiser, 2003*; *Schaap, 2011*; *Smith et al., 2019*) and many processes within complex multicellular organisms, for example, embryogenesis, tissue repair and cancer (*Weijer, 2009*; *Friedl and Gilmour, 2009*). Previous models have shown how cell collectives are able to integrate noisy information from the environment, for instance when moving up a shallow chemoattractant gradient. (*Marée et al., 1999*; *Szabó et al., 2006*; *Kabla, 2012*; *Szabó et al., 2010*; *Camley and Rappel, 2017*; *George et al., 2017*; *Camley, 2018*; *Varennes et al., 2017*).

We use the Cellular Potts Model (*Graner and Glazier, 1992*) (CPM) to study collective cell movement as an emergent driver of multicellularity during evolution. The CPM formalism is a spatially extended, mesoscopic description of cells which explicitly accounts for cell shape and size, and allows for a straightforward implementation various cellular processes within complex and potentially self-organised environments. We include four key elements: cells are placed in a seasonally changing environment that periodically introduces new resources at different locations, they can perform chemotaxis by sensing a chemoattractant produced by these resources, they reproduce depending on their proximity to resources and they can evolve their adhesion to other cells. Because the gradient generated by the resources is noisy and shallow, we find that individual cells follow the chemotactic signal very inefficiently. Instead, cells that adhere to each other within groups transfer information about the gradient in a self-organised manner, allowing for efficient chemotaxis in our model. We show that for longer seasons, this emergent property of cell groups is sufficient to select for high levels of adhesion and multicellularity, despite the fact that fitness is only defined at the cell level.

## Results

### Model setup
#### Cell model

We consider a population of $N$ cells that search for resources on a surface to be able to replicate. Cells are modelled with a 2D hybrid Cellular Potts Model (CPM) (*Graner and Glazier, 1992*; *Glazier and Graner, 1993*; *Daub and Merks, 2015*) on a square lattice of size $L^2 = 500 \times 500$ sites. The CPM formalism captures the fact that biological cells are dissipative objects with deformable boundaries. A cell consists of multiple adjacent lattice sites. The sites not occupied by cells are the medium, which contributes to determining the adhesive properties of a cell, but has no further properties. All the lattice sites belonging to one cell have the same identification number, different from that of any other cell or medium. Cell movement arises from stochastic fluctuations (extensions and retractions) of the cell boundaries. These fluctuations are generated by forces arising from cell size maintenance, adhesion and migration (explained below). We calculate these forces by minimising the corresponding energy function with the Metropolis algorithm (with a temperature-like parameter $T$ that scales the overall probability of membrane fluctuations). Lattice sites are updated in random order. In one Monte Carlo Step (MCS), $L^2$ lattice sites are updated.

To model cells as elastic and deformable objects, we assume that cell size - the number of lattice sites it is made up of - remains close to a preferred value $A_T$ equal for all cells (set to 50 lattice sites unless explicitly stated), and deviations are resisted with a stiffness parameter $\lambda$. Cells adhere to each other if they express matching ligands and receptors on their surface. Ligands and receptors are modelled as bit strings of length $\nu$ (*Figure 1a*), and are assumed to be expressed constitutively and uniformly on the membrane. Adhesion strength increases linearly with the number of complementary bits in the ligand and receptor. In the CPM, adhesion strength is expressed in terms of the interfacial energy $J_{c,c}$. For each pair of adjacent lattice sites belonging to different cells, the interfacial energy $J_{c,c}$ is calculated from the cells' ligands and receptors. A larger complementarity corresponds to lower values of $J_{c,c}$ (i.e. lower energy level in the bound state) and thus stronger binding. For cells adjacent to the medium, an additional cell-medium contact energy $J_{c,m}$ is calculated based on the similarity between part of their ligand bit string and an arbitrary target string. Cells adhere when cell–cell contact energy and medium-medium energy (equal to zero by definition) are lower than cell-medium contact energy: $(J_{m,m} + J_{c,c})/2 < J_{c,m}$. Cell adhesion can be characterised through

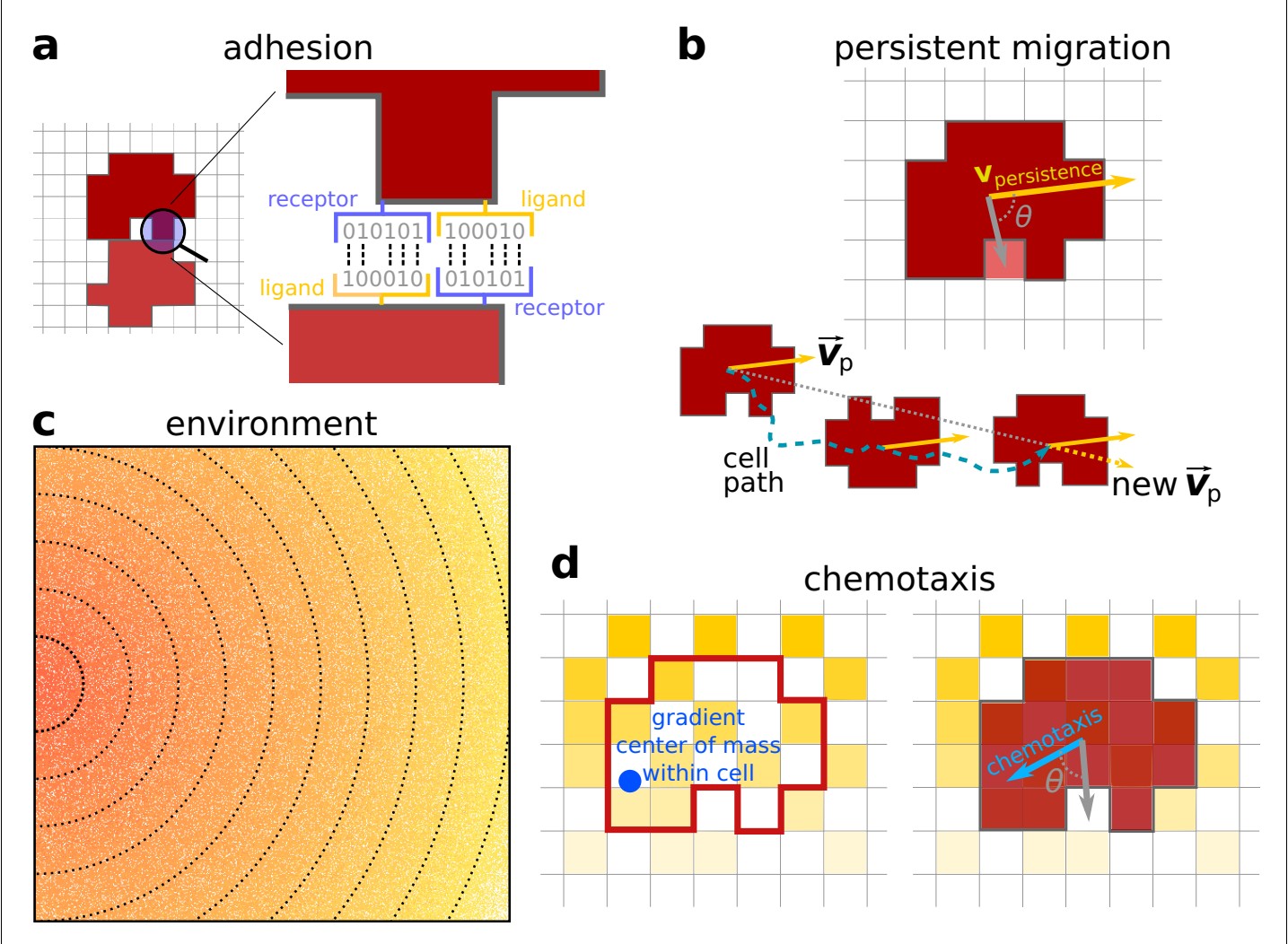

**Figure 1.** Model description. (a) Adhesion between two cells is mediated by receptors and ligands (represented by a bitstring, see *Hogeweg, 2000*). The receptor of one cell is matched to the ligand of the other cell and vice versa. The more complementary the receptors and ligands are, the lower the J values and the stronger the adhesion between the cells. (b) Persistent migration is implemented by endowing each cell with a preferred direction of motion $\vec{v}_p$. Every $\tau_p$ MCS, this direction is updated with a cell's actual direction of motion in that period. (c) The chemoattractant gradient in the lattice. The lines and colour indicate equal amounts of chemoattractant. Note the scattered empty lattice sites. (d) A cell can only sense the chemoattractant in the lattice sites that correspond to its own location. The cell will then move preferentially in the direction of perceived higher concentration, the chemotaxis vector. This vector points from the cell's centre of mass to the centre of mass of the chemoattractant detected by the cell (the blue dot).

the surface tension $\gamma = J_{c,m} - J_{c,c}/2$ (*Glazier and Graner, 1993*). Cells adhere when $\gamma > 0$ and disperse for $\gamma < 0$. Note that the value of $\gamma$ is due to a balance between $J_{c,c}$ and $J_{c,m}$, such that cells achieve higher surface tension either by increasing the number of complementary ligand-receptor pairs or by reducing the similarity of their ligands to the medium target string. Modelling ligands and receptors separately allows for sufficient variability of differential adhesion, without predetermining the J values between cells. For example, it allows for any combination of adhesion strengths between three (or more) cells.

During chemotaxis, eukaryotic cells repeatedly reorient the polymerisation of the actin cytoskeleton. This reorientation takes some time (*Ridley et al., 2003*), resulting in a migration pattern that is persistent over short time scales. We emulate this by combining a model of persistent migration (*Beltman et al., 2007*) with chemotaxis. Following (*Beltman et al., 2007*), persistent migration occurs through biasing cell membrane fluctuations towards the previous direction of motion of a cell

(*Figure 1b*). The strength of the bias is quantified by $\mu_p$, and the direction of motion is updated every $\tau_p$ MCS with the direction of actual cell displacement. This model of cell migration generates a persistent random walk (*Beltman et al., 2007*). Chemotaxis biases cell motion (with strength $\mu_\chi$) towards higher local concentrations of a chemoattractant (*Figure 1c,d*). We assume that this chemo-attractant is released at low concentration by resources present at one end of the grid, creating a shallow and noisy gradient over the grid (*Figure 1c*). For simplicity, we model a shallow, linear and noisy gradient decreasing from the source with slope $k_\chi$ (in unit of percent decrease over unit distance), and a heterogeneous substrate on which the chemoattractant may not attach (with probability $p_{\chi=0}$). Because of the noise in the gradient, the direction of cells' chemotaxis may be different from the correct direction of the gradient. We used this model setup to assess the properties of single-cell vs. collective migration.

## Evolutionary model

To explore the evolutionary dynamics of a population of cells, we seasonally change the location of the resources, and therewith the direction of the gradient, every $\tau_s$ MCS (*Figure 2*). Longer seasons (larger values of $\tau_s$) correspond to more persistent resources. During each season (i.e. one period of $\tau_s$ MCS) cells move due to chemotaxis and persistent migration. Depending on the ligands and receptors expressed on the cell surfaces, they may either adhere to one another or disperse from one another (*Figure 2a*). At the end of the season, cells are given a chance to divide, followed by a culling phase to keep the number of cells constant. To reflect the assumption that more nutrients are present at higher concentrations of the signal, the division probability is inversely proportional to the distance of the cell to the gradient peak and cells very close to the gradient peak may divide multiple times. Cells divide along their short axis to create two daughter cells (after *Hogeweg, 2000*), after which we let cells regrow to target size for 5 MCS. The daughter cells inherit mutated copies of the ligand and receptor, so that their adhesive properties can change with respect to the parent. This allows cells to evolve their adhesion strength. Cell size $A_T$, strength of chemotaxis $\mu_\chi$ and migration persistence $\mu_p$ do not evolve. After cell division, the population is brought back to $N$ cells by randomly culling cells, at which point the new season begins (*Figure 2b*). Note that we do not include cell dispersal after replication, therefore related cells remain close at the beginning of the new season. Simulations last 400 seasons (i.e. $400 \times \tau_s$ MCS), which is sufficient to reach evolutionary steady state under all conditions.

We do not select for multicellularity directly: the fitness function rewards cells for their proximity to resources, and we do not explicitly incorporate a fitness advantage or disadvantage for the

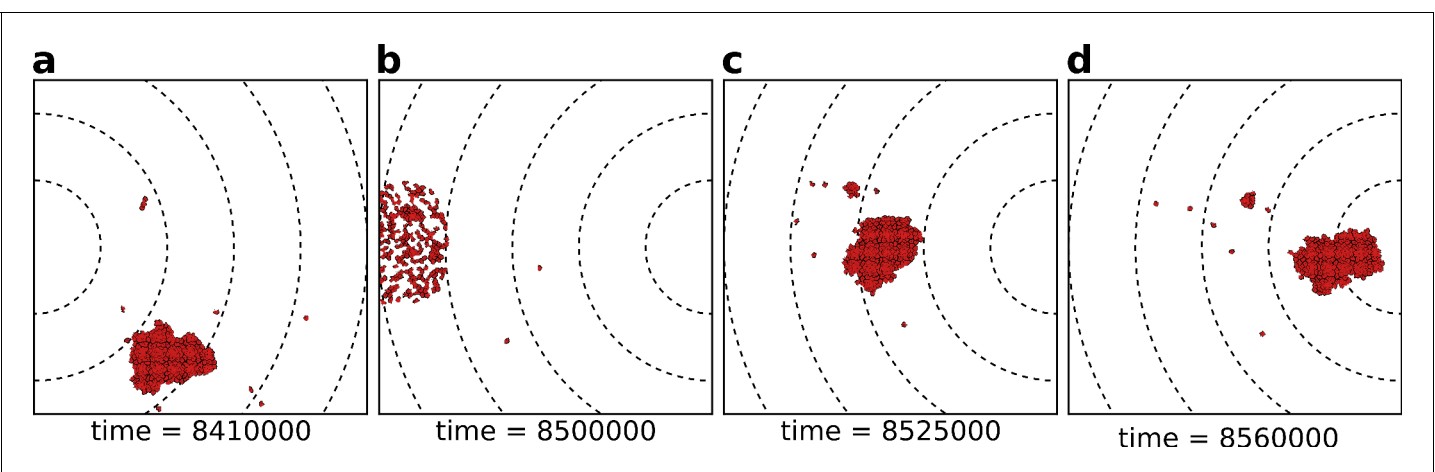

**Figure 2.** The eco-evolutionary setup of the model. (a) A population of $N = 200$ cells moves by chemotaxis towards the peak of the gradient, which in this season is located at the left boundary of the grid. (b) At the end of the season, cells divide, the population excess is killed randomly, and the direction of the chemotactic signal is changed, after which the new season begins (c, d). The snapshots are taken at the indicated time points from a simulation where a season lasts $\tau_s = 100 \times 10^3$ MCS. Dashed lines in the snapshots are gradient isoclines.

multicellular state. Therefore, multicellular clusters (*Figure 2a,c,d*) can arise only because they perform an emergent task that single cells cannot perform.

See *Table 1* for parameter values, and Materials and methods Section for the details of the model and parametrisation.

## Strongly adhering cells perform efficient collective chemotaxis

We first assessed how well groups of cells with different adhesion strengths could reach the source of the chemotactic signal. We placed a connected cluster of cells on one side of the lattice, opposite to the location of the gradient peak. We then recorded their travel distance over a fixed amount of time and compare it to the travel distance of single cells (i.e. from simulations with only one cell), by measuring both the position of the centre of mass of the group (*Figure 3a*) and the position of the cell closest to the peak of the gradient (*Figure 3b*). Single cells perform chemotaxis inefficiently (*Video 1*), whereas a group of adhering cells migrates up the same gradient more accurately (*Figure 3a*, $\gamma > 0$): the centre of mass of this group takes much less time than single cells do to reach the peak of the gradient (*Video 2*). Groups of cells can also perform collective chemotaxis when they do not adhere, and when they do not have a preference for medium or cells, although with lower efficiency in both cases (*Figure 3a*, respectively $\gamma < 0$ and $\gamma = 0$). Chemotaxis is inefficient, because these cells tend to lose contact from one another (*Video 3*) and once isolated they behave like those from simulations with one cell (*Figure 3a,b* 'one cell'). Single cells also show large variance between different simulations (*Figure 3b*). While cell clusters perform chemotaxis efficiently only when cells adhere, the speed of the cell closest to the peak of the gradient is roughly the same regardless of adhesion strength (*Figure 3b*). Thus, in a non-adhering population some cells reach the peak of the gradient almost as quickly as an adhering cluster does.

Adhering cells have large chemotactic persistence - as shown by the super-linear shape of the Mean Square Displacement (MSD) plot (*Figure 3c*, $\gamma = 6$) and by a diffusive exponent consistently

**Table 1.** Parameters.

| Parameter | Explanation | Values |
|---|---|---|
| $L^2$ | lattice size | $500 \times 500$ lattice sites |
| $T$ | Boltzmann temperature | 16 AUE |
| $\lambda$ | cell stiffness | 5.0 AUE/[lattice site]$^2$ |
| $A_T$ | cell targetarea | 50 lattice sites |
| *Cell adhesion* | | |
| $J_\alpha$ | minimum J value between cells | 4 AUE/[lattice site length] |
| $J'_\alpha$ | minimum J value between cell and medium | 8 AUE/[lattice site length] |
| $\nu$ | length of receptor and ligand bitstring | 24 bits |
| $\nu'$ | length ligand bitstring for medium adhesion | six bits |
| *Cell migration and chemotaxis* | | |
| $\mu_p$ | strength of persistent migration | 3.0 AUE |
| $\tau_p$ | duration of persistence vector | 50 MCS |
| $\mu_\chi$ | strength of chemotaxis | 1.0 AUE |
| $k_\chi$ | scaling factor chemoattractant gradient | 1.0 molecules/[lattice site length] |
| $p_{\chi=0}$ | probability of zero value ('hole') in gradient | 0.1 [lattice site]$^{-1}$ |
| *Evolution* | | |
| $N$ | population size | 200 cells |
| $\tau_s$ | duration of season | $5 \times 10^3$ - $150 \times 10^3$ MCS |
| $h_d$ | distance from gradient peak where fitness is $\frac{1}{2}$ | 50 [lattice site length] |
| $\mu_{R,I}$ | receptor and ligand mutation probability | 0.01 per bit, per replication |

AUE: Arbitrary Units of Energy (see Hamiltonian in Model Section); lattice site: unit of area; lattice site length: unit of distance; MCS: Monte Carlo Step (unit of time).

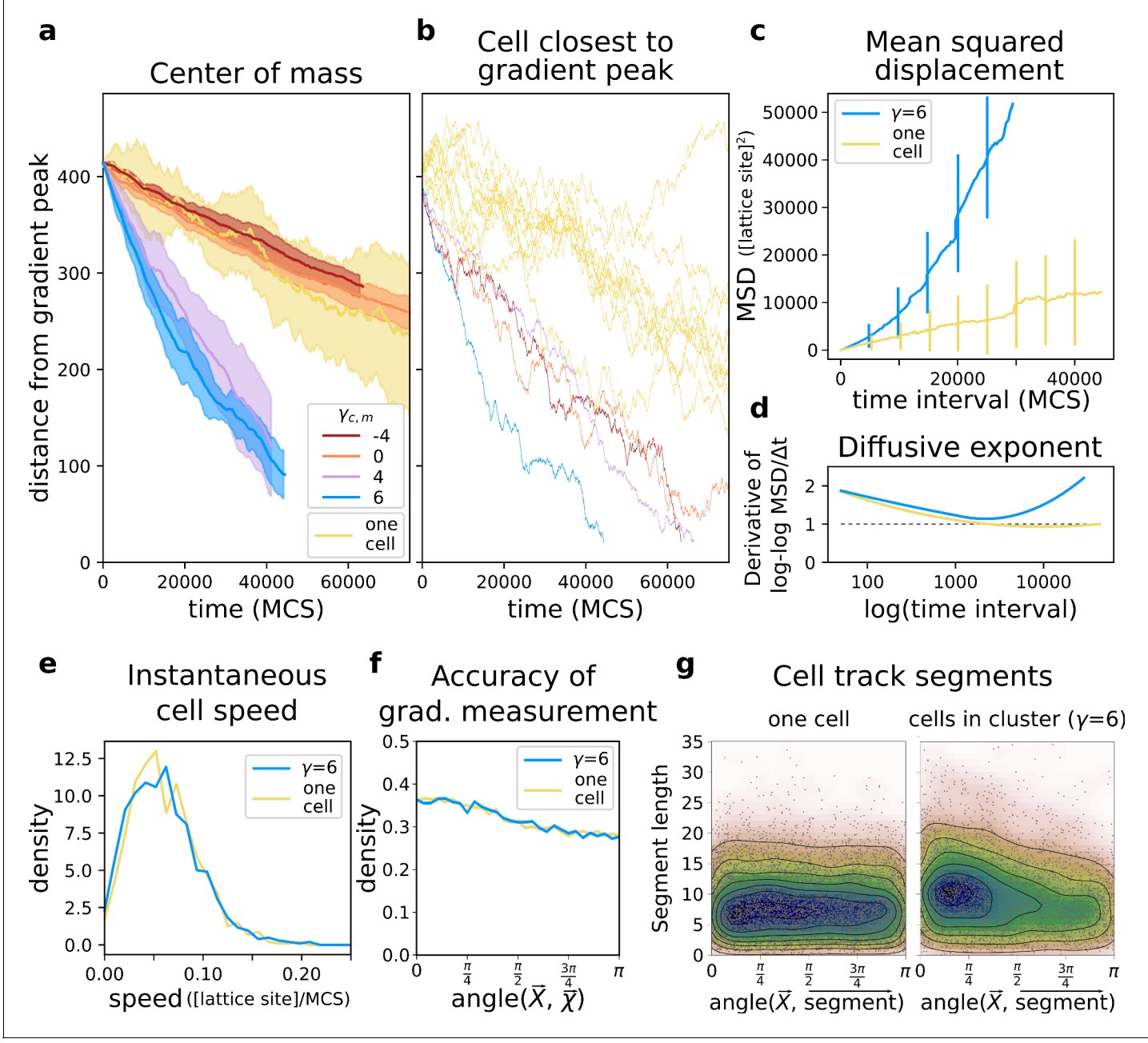

**Figure 3.** A group of cells performs chemotaxis efficiently in a noisy shallow gradient. (a) Distance of the centre of mass of $N = 50$ cells from the peak of the gradient as a function of time, for different values of $\gamma \in [-4, 6]$ (five independent runs for each value), together with the average position of 10 isolated cells (i.e. from simulations with only one cell). (b) The position of the cell closest to the gradient origin as a function of time (taken from the same simulations as in a), and the positions of 10 individual cells (whose average generates the corresponding plot in a). (c) Mean square displacement (MSD) per time interval for two datasets each with 50 simulations of either single cells or clusters of strongly adhering cells ($N = 50$, $\gamma = 6$), in which case we extracted one cell per simulation. These data sets were also used for the following plots. (d) Diffusive exponent extracted from the MSD plot, obtained from the log-log transformed MSD plots by fitting a smoothing function and taking its derivative (Appendix 1.1). (e) Distribution of instantaneous cell speeds (f) Distribution of angles between cells' measurement of the gradient $\vec{\chi}$, and the actual direction of the gradient peak $\vec{X}$, as measured from the position of the cell. (g) The length of straight segments in cell tracks vs. their angle with the actual gradient direction. Each point represents one segment of a cell's trajectory. To extract these straight segments a simple algorithm was used (Appendix 1.8). Contour lines indicate density of data points.

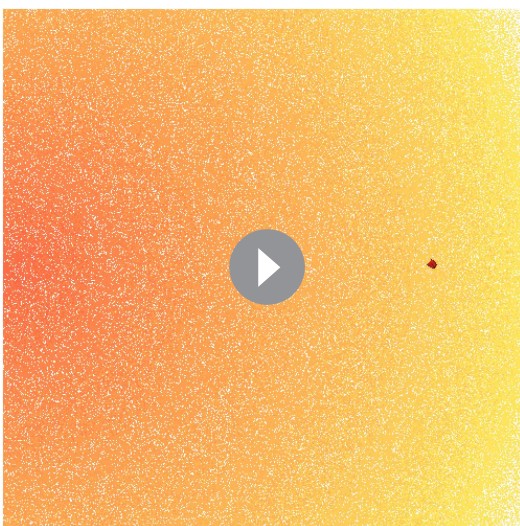

**Video 1.** Inefficient chemotaxis of a single cell.
https://elifesciences.org/articles/56349#video1

larger than 1 (**Figure 3d**; the diffusive exponent is obtained as the derivative of the log-log transformed MSD/time curve, see Appendix 1.1). Instead, the MSD of a single cell (**Figure 3c**, one cell) is approximately linear and its diffusive exponent tends to 1, indicating that cells' movement is much more dominated by diffusion. Interestingly, there is no difference in the instantaneous speed of cells when they are in a cluster or when they are alone (**Figure 3e**), so the higher rate of displacement of a group of adhering cells is only due to larger persistence in the direction of motion. **Figure 4** shows the movement of a cluster of strongly adhering cells ($\gamma = 6$) compared to the movement of a single cell, over the typical setup of the simulation system. Although the cluster moves straight towards the source of the gradient, individual cells follow noisy trajectories.

A possible explanation for collective chemotaxis is that a cluster averages individual cells' polarisation, leading to a linear relationship between the accuracy of chemotaxis and the number of cells in the cluster (**Varennes et al., 2017**). Instead, we found that cluster speed saturates quickly with the number of cells, at a smaller speed than that of individual cells (cf. Appendix 1.2 with **Figure 3e**). We conclude that individual contributions to cluster chemotaxis are not simply averaged. Therefore, we look at how cells self-organise to understand how collective chemotaxis comes about.

Through persistent migration, a cell pushes other cells within an adhering cluster, and is pushed by them. The resulting forces are resolved when cells align and form streams within the cluster (see **Video 4**). These streams are persistent over a much longer time scale than a cell's persistence $\tau_p = 50$ MCS (since the video frame rate is 50 MCS and streams are visible over multiple frames). Through streaming, these small clusters generate extensions, retractions and rotations (**Video 4**), so that the entire cluster visually resembles a single amoeboid cell (**Video 2**). This behaviour is not influenced by the presence of the chemotactic signal, since the flow field is identical when the chemotactic signal is removed (Appendix 1.3). Thus, the effect of persistent migration is to align the direction of motion of the cells in a cluster. This in turn speeds up collective chemotaxis, as cell streams preferentially align towards the direction of the gradient, although aligning is not strictly required for chemotaxis (Appendix 1.4). Clusters perform chemotaxis faster than individual cells over a large range of values for persistent migration strength $\mu_p$ and chemotactic strength $\mu_\chi$ (Appendix 1.5 and Appendix 1.6), with larger $\mu_p$ increasing collective chemotaxis speed (and to a lesser extent individual chemotaxis speed) more than $\mu_\chi$. Because larger cells perceive a larger area of the chemotactic signal, chemotactic migration improves with cell size (Appendix 1.7).

We calculated the deviation of each individual cell's measurement of the gradient as the angle $\theta(\vec{X}, \vec{\chi})$ between the true direction of the

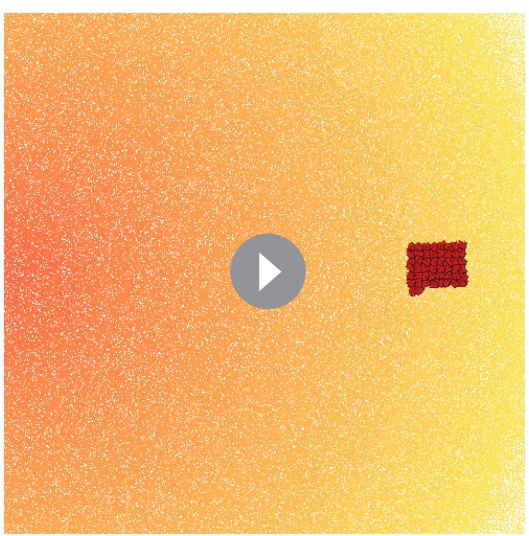

**Video 2.** Chemotaxis of a cluster of adhering cells. All cells have the same colour to show how the migration of the cluster as a whole resembles that of an amoeba.
https://elifesciences.org/articles/56349#video2

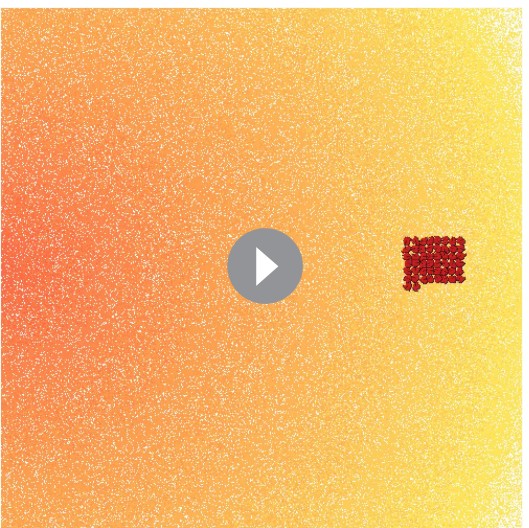

**Video 3.** Inefficient chemotaxis of a cluster of non-adhering cells.
https://elifesciences.org/articles/56349#video3

gradient $\vec{X}$ and the direction of the gradient locally measured by the cells $\vec{\chi}$ (so that $\theta(\vec{X}, \vec{\chi}) = 0$ is a perfect measure). We found that the measurements of individual cells deviate significantly from the true direction of the gradient (*Figure 3f*). Despite this, they are carried in the right direction by the other cells. To assess how cells in a cluster alter each others' (short-time-scale) trajectories we extracted the straight segments from the cell tracks and assessed both the length of these segments and their orientation with respect to the gradient source (Appendix 1.8). We find that cells in a cluster tend to migrate for longer in straight lines, and that these straight lines are also more likely to be oriented towards the source of the gradient (*Figure 3g*). For single cells, there is no such bias.

In conclusion, cluster organisation emerges from cells altering each others' paths by exerting pushing and pulling forces through their persistent migration, which in turn results in efficient collective chemotaxis.

## The evolution of uni- or multicellular strategies depends on season duration

The emergence of reliable chemotactic behaviour in adhering cell clusters suggests an evolutionary path to multicellularity: a population of cells may aggregate if collective chemotaxis allows cells to find resources more reliably. While cells could improve their ability to sense the gradient individually by becoming bigger, there are many factors that restrict cell size, such as the complexity of the metabolism and cellular mechanisms such as cell division (*Björklund and Marguerat, 2017*; *Marshall et al., 2012*). We therefore assume that cell size is fixed, and we let cells evolve adhesion -

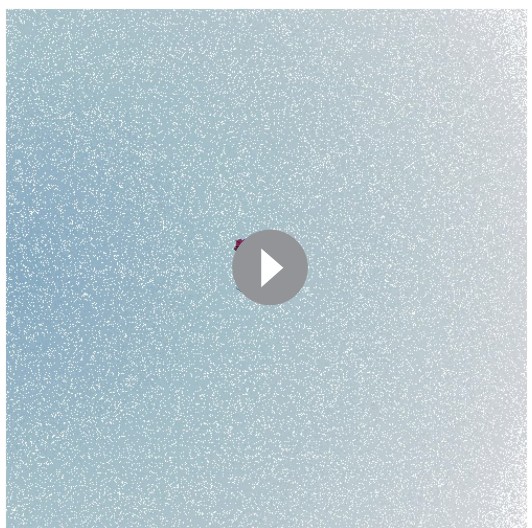

**Video 4.** The same cluster of adhering cells. Cell colour indicates the direction of migration, to emphasise the streaming dynamics within the cluster.
https://elifesciences.org/articles/56349#video4

that is, the receptor and ligands expressed by the cells - in response to a seasonally changing environment, where the gradient is generated by a volatile resource that periodically changes position. Cells closer to the peak of the gradient have a higher chance to reproduce at the end of the season, and related cells remain close to each other at the beginning of the new season (there is no cell dispersal phase, see also model setup and Materials and methods). The receptors and ligands of the initial population are chosen such that cells neither adhere to one another nor disperse from one another ($\gamma = 0$).

When the season lasts $\tau = 100 \times 10^3$ MCS, the average adhesion between cells readily increases after only few generations (*Figure 5a*): $J_{cell,cell}$ decreases and $J_{cell,medium}$ increases (see also *Video 5* and *Figure 2* for snapshots). At evolutionary steady state, all cells adhere strongly and with roughly the same energy to one another (Appendix 2.1). *Figure 5b* shows that two evolutionary steady states are possible, depending on the duration of the season $\tau_s$. For $\tau_s < 20 \times 10^3$

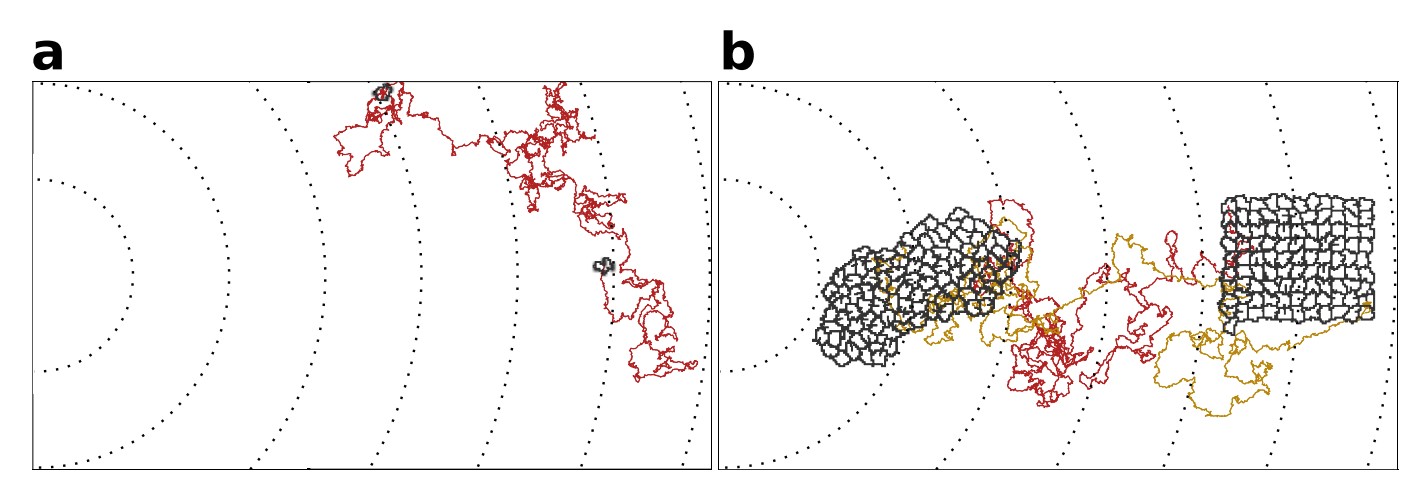

**Figure 4.** Indivdual cell trajectories are noisy, also within a cluster. (**a**) The movement of a single cell. (**b**) Typical movement of a cluster of strongly adhering cells, and of the cells inside the cluster. Cells are placed on the right of the field and move towards higher concentration of the gradient (to the left of the field). Dashed lines are gradient isoclines.

MCS, cells evolve to become unicellular, as cell–cell interactions are characterised by strong repulsion ($\gamma < 0$). *Figure 5c* suggests that by selecting for $\gamma < 0$ cells disperse efficiently throughout the grid. Although non-adhering cells follow the chemotactic signal only weakly, the spreading over the course of multiple seasons ensures that at least some cells end up close to the source of the gradient at the end of the season (*Video 6*). In contrast, a cluster of adhering cells is at disadvantage when seasons are short because it does not have enough time to reach the source of the chemotactic signal. Over the course of multiple seasons, an adhering cluster ends up in the centre of the lattice (*Video 7*) and all its composing cells have the same (low) fitness. Furthermore, the connectedness of a cluster of adhering cells is locally disrupted when excess cells are culled between seasons (*Figure 2b*), which briefly reduces the efficiency of collective migration. Because this phase is short-lived - cells reconnect within 2000 MCS - we expect that culling plays a minor role in the evolutionary outcome of the system. For $\tau_s > 40 \times 10^3$ MCS, cells evolve to adhere to one another, i.e. $\gamma > 0$ (see *Figure 5c* for a snapshot). When seasons are sufficiently long, clusters of adhering cells have enough time to reach the source of the gradient. At this point, the fitness of cells within a cluster outweighs that of non-adhering cells, because clustering increases the chances of reaching the peak of the gradient. Finally, for intermediate season duration, $20 \times 10^3 \leq \tau_s \leq 40 \times 10^3$ MCS, both repulsion and adhesion are evolutionary (meta) stable strategies, and the outcome of the simulation depends on the initial value of $\gamma$ (for $\tau_s = 20 \times 10^3$ MCS, the steady state with $\gamma > 0$ is very weakly stable).

Because different values for migration parameters affect collective chemotaxis speed, we checked that the evolution of multicellularity is qualitatively robust to changes in the values of persistent migration strength $\mu_p$ and chemotactic strength $\mu_\chi$ (respectively in Appendix 2.2 and Appendix 2.3). Results are also robust to changes in gradient shape (assuming that resources are

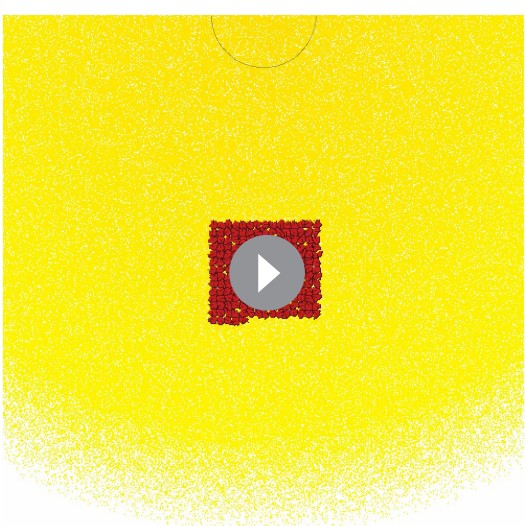

**Video 5.** Video of an evolutionary simulation, starting with neutrally adhering cells ($\gamma = 0$). The season changes every $100 * 10^3$ MCS.
https://elifesciences.org/articles/56349#video5

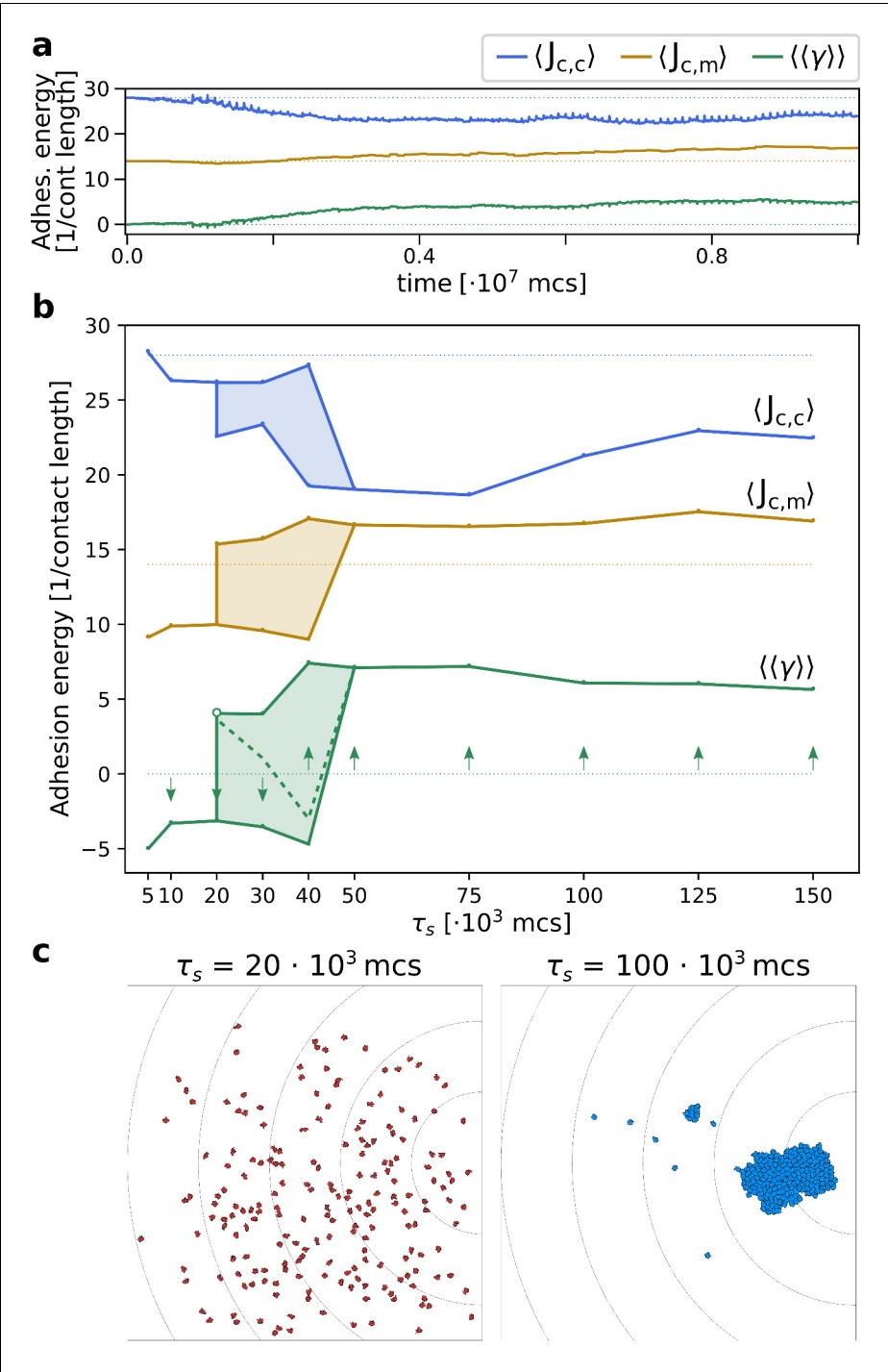

**Figure 5.** The evolution of multicellularity. (a) Multicellularity ($\gamma > 0$) rapidly evolves in a population of $N = 200$ cells with $\tau_s = 10^5$. (b) Multicellularity only evolves when seasons are sufficiently long $\tau_s \geq 50 * 10^3$; unicellular strategies evolve when seasons are short $\tau_s \leq 10 * 10^3$, and both strategies are viable depending on initial conditions for intermediate values of $\tau_s$. The dashed line indicates the separatrix between the basins of attraction of the two evolutionary steady states; it is estimated as the mid-point where evolutionary simulations with consecutive initial values of $\gamma \in \{-6, -4, -2, 0, 2, 4, 6\}$ evolve to alternative steady states. In both panes, $\langle\langle\gamma\rangle\rangle$ is estimated as $\langle J_{c,m}\rangle - \langle J_{c,c}\rangle / 2$, where $\langle J_{c,c}\rangle$ and $\langle J_{c,m}\rangle$ are calculated from the $J_{c,c}$ and $J_{c,m}$ extracted from the system at evolutionary steady state. The initial J values, indicated by the dotted lines, are such that $\gamma = 0$. (c) Snapshots of the spatial distribution of the population at evolutionary steady state for $\tau_s = 20 \cdot 10^3$ and $\tau_s = 100 \cdot 10^3$ MCS.

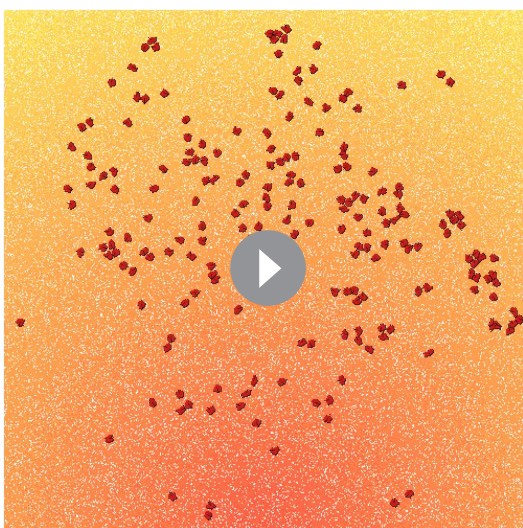

**Video 6.** Over time a population of non-adhering cells spread throughout the lattice, when seasons are short. The season changes every $10 * 10^3$ MCS. For all cells $\gamma = -4$. Mutation rate is set to zero to emphasise the spatial population dynamics.
https://elifesciences.org/articles/56349#video6

located over an entire side of the lattice, we tested a gradient with straight isoclines in Appendix 2.4) and to steeper, noiseless gradients (Appendix 2.5). Furthermore, the evolution of multicellularity does not depend on the precise mechanism for collective chemotaxis. To show this, we relax the assumption that individual cells sense the gradient by implementing a recently proposed mechanism of emergent collective chemotaxis that relies only on concentration sensing (*Camley et al., 2016*). Following *Camley et al., 2016*, we assume that cell polarisation is inhibited at the sites of cell–cell contact (a phenomenon called contact inhibition of locomotion, see *Mayor and Carmona-Fontaine, 2010* for a review), and that the magnitude of their polarisation is proportional to the concentration - not the gradient - of the chemoattractant. In Appendix 3.1, we show that results are robust to this modification of the chemotaxis mechanism.

## Interference competition between unicellular and multicellular strategies causes evolutionary bi-stability

We next investigated what causes the evolutionary bi-stability in adhesion strategies for season duration $20 \times 10^3 \leq \tau_s \leq 40 \times 10^3$ MCS. We performed competition experiments between two populations of cells, one adhering ($\gamma = 6$) and one non-adhering ($\gamma = -4$), to determine whether a strategy can invade in a population of cells using the other strategy. We simulated non-adhering mutants invading a resident population of adhering cells by placing a large cluster of adhering cells in front of a small group of non-adhering ones (*Figure 6a*), and conversely, a small cluster of adhering cells invading a large group of non-adhering cells (*Figure 6b*). This initial configuration is analogous to the beginning of a season in the evolutionary experiments, as mutants are in small numbers and furthest away from the new peak because they are likely born from cells that replicate most, that is, those closest to the previous location of the peak. In both cases, after $30 \times 10^3$ MCS, the resident population physically excludes the invading one from the path to resources, and thus the distance travelled by the invading population is limited. This shows that the adhesion energy of the resident population (whether cells adhere or not) determines the outcome of the invasion (for the values of $\tau_s$ where we find evolutionary bistability). We also considered a scenario where a whole population - rather than few mutants - invades another with the opposite strategy. We studied the spatial competition dynamics of two clusters of equal size ($N = 100$ cells) when adhering cells are positioned in front of the non-adhering ones (*Figure 6c*), and when the position of the two

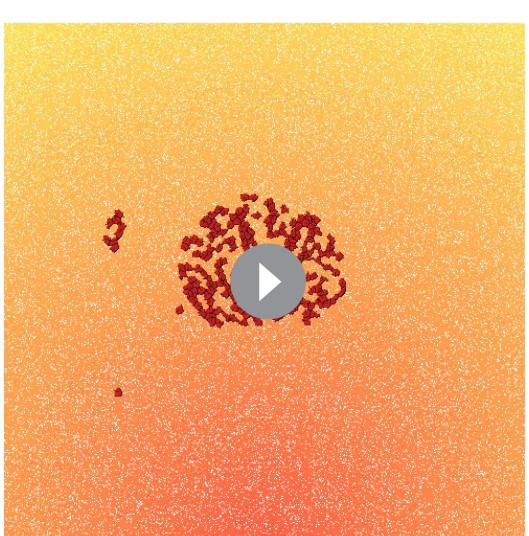

**Video 7.** Over time a population of adhering cells ends up in the centre of the lattice when seasons are short. The season changes every $10 * 10^3$ MCS. For all cells $\gamma = 6$. Mutation rate is set to zero.
https://elifesciences.org/articles/56349#video7

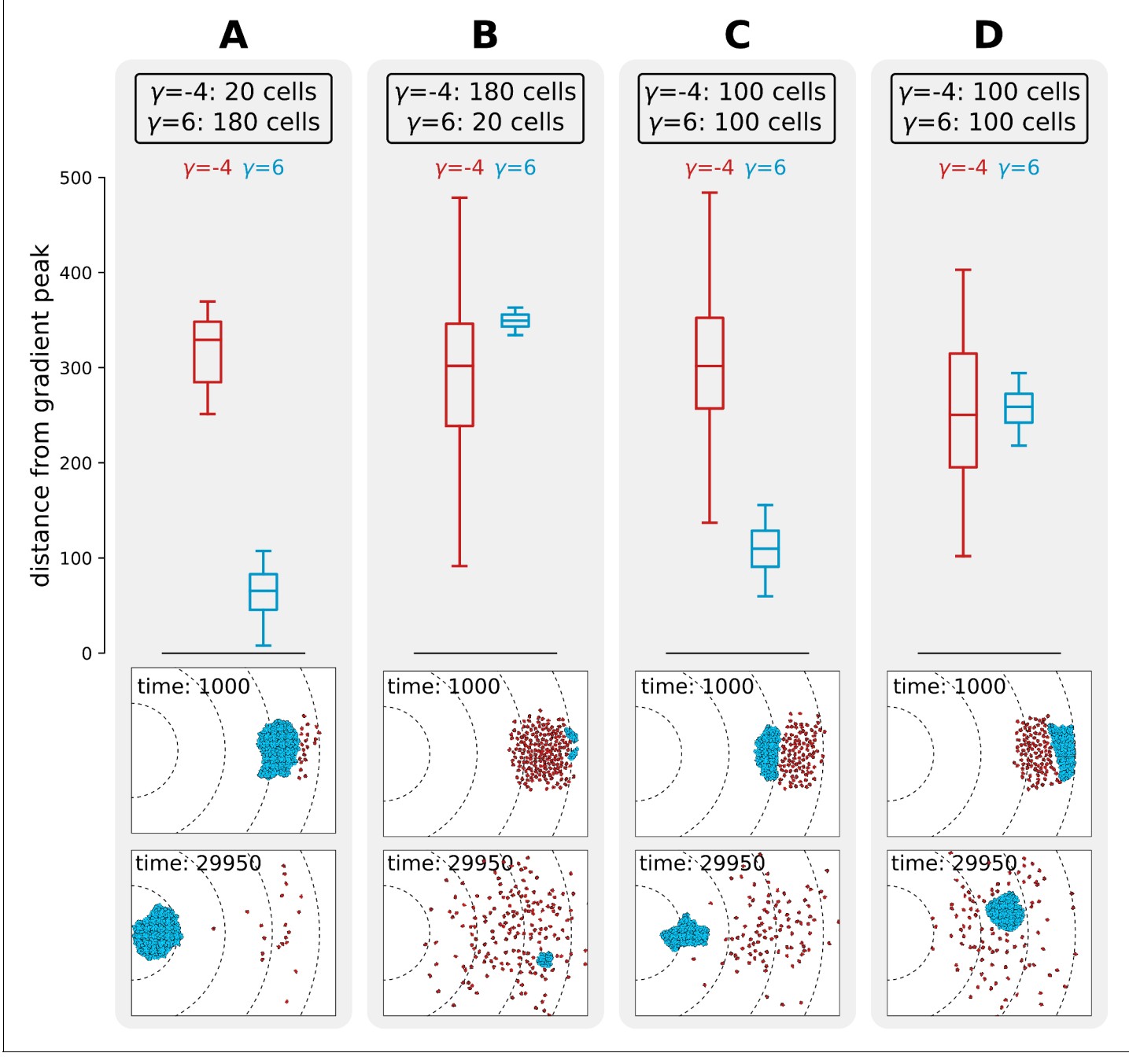

**Figure 6.** Interference competition between adhering and non-adhering cells explains evolutionary bistability. We let a simulation run for $\tau_s = 30 \times 10^3$ MCS and then record the distance from the peak of the gradient, for two different populations of cells - one non-adhering (in red, $\gamma = -4$) and one adhering (in blue, $\gamma = 6$), for different initial conditions. The snapshots underneath are the initial and final spatial configurations of the cells on the grid. (A) 180 adhering and 20 non-adhering cells, placed so that the adhering cells are closer to the source of the gradient; (B) 20 adhering and 180 non-adhering cells, placed so that the non-adhering cells are closer to the source of the gradient; (C) 100 adhering and 100 non-adhering cells, placed so that the adhesive ones are closer to the source of the gradient; (D) 100 adhering and 100 non-adhering cells, placed so that the non-adhering cells are closer to the source of the gradient. Dashed lines in the snapshots are gradient isoclines.

clusters is swapped (*Figure 6d*). The distance to the peak after $30 \times 10^3$ MCS of a cluster of adhering cells is larger (i.e. their fitness is smaller) if they are hindered by a population of non-adhering cells in front of them. Taken together, these results show that there is interference competition (i.e. direct competition due to displacement) between populations of cells with different strategies. In

the evolutionary experiments, mutants with a slightly different strategy are generated during reproduction at the end of each season and interference competition continually prevents their successful invasion for intermediate season duration. This explains why the two strategies are meta-stable. This result may also provide a simple explanation for the fact that many unicellular organisms do not evolve multicellularity despite possessing the necessary adhesion proteins. Moreover, evolutionary bi-stability protects the multicellular strategy from evolutionary reversal to unicellularity over a large range of environmental conditions.

## Multicellularity and the cost of adhesion

So far, we showed that the evolutionary benefit of uni- or multi-cellular strategies is indirect, as it is mediated by the fittest form of self-organisation for a given season duration. For simplicity, we did not incorporate any cost to evolving multicellularity. However, evolving multicellular organisms may incur fitness costs that are not present at the unicellular level (*Rebolleda-Gómez and Travisano, 2018*; *Rainey and Rainey, 2003*; *Ratcliff et al., 2012*; *Yokota and Sterner, 2011*; *Kapsetaki and West, 2019*). We incorporated costs in our system by assuming that cells spend energy to maintain their bonds with other cells, with a cost $c_m$ (per unit of cell boundary, per MCS). This metabolic cost accumulates over time when cells are in contact with one another, and translates into a fitness penalty at the end of the season for cells that spent more time in contact with others. Costs range from $c_m = 0$, the cost-free model presented so far, to $c_m = 1$ (the maximum cost) which zeroes the fitness of a cell that spent the entire season completely surrounded by other cells. Multicellularity evolved for sufficiently long seasons when costs were not too high ($c_m \leq 0.5$), with larger costs shifting the transition to multicellularity to longer seasons, while only the uni-cellular strategy evolved when costs were high ($c_m = 0.75$, for the season duration we tested; Appendix 4.1).

## Discussion

We demonstrated that undifferentiated multicellularity can evolve in a cell-based model as a byproduct of an emergent collective integration of spatial cues. Previous computational models have shown that multicellularity can be selected by reducing the death rate of cells in a cluster (*Staps et al., 2019*; *Pichugin et al., 2017*), through social interaction (*Garcia and De Monte, 2013*; *Joshi et al., 2017*), by incorporating trade-offs between fitness and functional specialisation (*Ispolatov et al., 2012*), or by allowing cells to exclude non-cooperating cells (*Pfeiffer and Bonhoeffer, 2003*). In these studies, direct selection for forming groups is incorporated by conferring higher fitness to the members of a cluster.

Earlier work found that multicellular structuring can emerge without direct selection when cells are destabilised by their internal molecular dynamics (e.g. the cell cycle) (*Furusawa and Kaneko, 2002*), or because of a toxic external environment (*Duran-Nebreda et al., 2016*). In both cases, cell differentiation stabilises cell growth and arises as a consequence of physiological or metabolic trade-offs. With our model setup, we show that division of labour - although important - is not a strict requirement for emergent aggregation. Nevertheless, our work bears some similarity with these models because we do not explicitly incorporate a fitness benefit for being in a group: selection acts on individual cells only on the basis of how close they are to the source of the gradient, regardless of migration strategy. Thus the fitness function does not dictate which evolutionary strategy, that is, uni- or multi-cellularity, should be followed.

A limitation of the current model is that cells have a narrow set of possibilities for adapting to the environment, as the only mutable traits are their ligands and receptors. Therefore, their adaptation to the environment is solely mediated by their adhesion to one another and selection for multicellularity can only occur because adhering clusters always perform chemotaxis better than individual cells. Despite the advantage of clusters over individuals, an alternative strategy can evolve that does not rely on collective behaviour. This uni-cellular strategy evolves because non-adhering cells disperse throughout the field over multiple seasons. By chance - and aided by inefficient chemotaxis - some cells will be located near the peak of the gradient at the end of each season. When seasons change rapidly, a multicellular cluster does not have the time to reach the peak of the gradient. It is therefore at disadvantage over cells evolving a unicellular strategy. This further illustrates that the selection pressure to become multicellular emerges from the structure of the environment in our model, rather than being an explicit part of the fitness function. Whichever evolutionary strategy

maximises fitness, be it multi- or uni- cellularity, will evolve within the (limited) complexity of the model.

A second limitation of the model is that resources are modelled only implicitly - through the chemoattractant gradient they generate and through season duration, i.e. how long they persist. The precise seasonality of these resources might be realistic if resources are deposited in the system by periodic phenomena (e.g. tides, or daily and yearly cycles), whereas other types of resources might be more stochastic (such as preys). However, if the stochasticity of resources is not too extreme, we expect that evolution converges to the average resource duration.

In many ways, the evolution of multicellularity can be compared to the evolution of collective dynamics. Previous studies on the evolution of herding behaviour showed that aggregating strategies can also evolve in response to highly clumped food even though the pack explores the space slowly and inefficiently before finding food (*Wood and Ackland, 2007*). When gradient sensing and social behaviour are both costly, a combination of strategies evolves in response to selection for distance travelled (*Guttal and Couzin, 2010*). Some individuals pay the cost for actively sensing the gradient, while others invest in social behaviour to move towards others and align their direction of motion with them, leading to the formation of migrating herds (*Guttal and Couzin, 2010*). These models of collective migration represent individuals as active particles, which is similar to the behaviour of our cells. However, group movement requires an explicit rule for alignment, whereas in our model it emerges naturally from interactions between deformable cells. Modelling cells with an explicit shape and size (including both CPM and, we expect, self-propelled particles) allows for spatial self-organisation and can generate interesting ecological dynamics, such as interference competition between the unicellular and multicellular search strategies. The ensuing evolutionary bi-stability stabilises unicellularity despite these cells possessing the surface protein toolkit to adhere to each other, and prevents multicellular organisation from evolutionary reversal into single cells (over a range of environmental conditions). The 'automatic' outcome of spatial self-organisation provides an initial, non-genetic robustness, which can be further stabilised by later adaptations (*Libby et al., 2016*).

In our model, cells retain their spatial distribution between seasons. This reinforces spatial self-organisation, and consequently bistability, because genetically similar cells remain close to one another. However, we expect bistability also if cells were dispersed between seasons: few adhering cells scattered in a cloud of non-adhering ones would not be likely to meet (and collectively chemotax) if seasons are short. In contrast, a large number of adhering cells would meet frequently after scattering and thus displace non-adhering cells in their march towards the peak of the gradient. This suggests that the two strategies are not mutually invadable over some intermediate season length, hence bistability.

The driver for the evolution of adhesion in our model is collective chemotaxis. This is reminiscent of the aggregate phase of the life cycle of *Dictyostelium discoideum* (*Schaap, 2011*), in that a cluster of cells moves directionally as a unit following light or temperature, while individual cells inefficiently identify the correct direction of motion (*Miura and Siegert, 2000*). There are some important differences between our model and *D. discoideum*, however. Information about the direction of the gradient is transmitted mechanically within cell clusters in our model. In *D. discoideum* photo- and thermo-taxis are coordinated by waves of cAMP secretion that travel through the slug. The lack of extra chemical cues to organise movement within a cell cluster in our model makes for a simpler scenario without large-scale transmission of information throughout the aggregate. Nevertheless, computational modelling has shown that long-range chemical signalling coupled to cells' differential adhesion suffices to reproduce *D. discoideum*'s migration (*Marée et al., 1999*; *Marée and Hogeweg, 2001*). Another important difference between our model and *D. discoideum* is the absence of dispersal at the end of the life cycle in our model. In *D. discoideum*, the slug transforms into a fruiting body at the top of a stalk of terminally differentiated cells. Extending the current model with the evolution of dispersal would enrich our understanding of *D. discoideum* evolution towards partial multicellularity.

Our model of collective movement is an example of the 'many wrongs' principle (*Simons, 2004*): the direction error of each cell is corrected by the interactions with the other cells in the cluster. However, in our model there is no explicit mechanism for transferring gradient information between cells. Therefore our results differ from previous work on rigid clusters of cells, where cells' polarisation towards the perceived gradient translates linearly and instantaneously to cluster movements

(*Camley et al., 2016*; *Varennes et al., 2017*). In models where cells readily exchange neighbours, simple rules for cell adhesion and migration led to self-organisation of cells into highly persistent, migrating tissue with emergent global polarity (*Smeets et al., 2016*) (earlier observed also at large cell density without adhesion rules [*Beltman et al., 2007*; *Szabó et al., 2006*]). Similarly, in our model, cells convey gradient information through such emergent collective streaming, which becomes biased towards the (weak) chemotactic signal. However, we expect that the evolutionary results described here are independent of the particular cell model choice, or the mechanism for chemotaxis provided that cells were able to polarise or move also in the absence of other cells. Indeed, we found similar results with an alternative model of collective chemotaxis (*Camley et al., 2016*) in which individual cells do not sense the gradient.

We opted for a computational cell-based model - the Cellular Potts Model - because it allowed us to explore the spatial interactions of cells, and because it enabled straightforward implementation of the evolvable receptor-ligand system. The visual nature of our results may guide the future development of analytical approaches to generalize the results of this work. For instance, analytical work may provide a more detailed explanation of the 'many wrongs' principle for a cell cluster in which cells are highly motile and change their neighbours often, in which case positional information is transmitted by pulling and pushing on each other. Moreover, the simplicity of our model setup makes our results easily testable in vitro.

The importance of a bottom-up approach to study the evolution of multicellularity has been repeatedly emphasised (*van Gestel and Tarnita, 2017*; *De Monte and Rainey, 2014*), and a broader understanding of cells self-organisation and evolution may have applications to clinically relevant multiscale evolutionary problems, such as the evolution of collective metastatic migration of cancer cells (*Coffey, 1998*; *Stuelten et al., 2018*; *Disanza et al., 2019*; *Lacina et al., 2019*). Our work highlights that the properties of single cells emergently give rise to novel properties of cell clusters. These novel properties - in a downward causative direction - generate the selection pressure to form the first undifferentiated multicellular groups.

## Materials and methods

We model an evolving population of cells that migrate and perform chemotaxis on a 2-dimensional lattice. Cell–cell interactions and movements are modelled with the Cellular Potts Model (CPM) (*Graner and Glazier, 1992*; *Glazier and Graner, 1993*) and simulated with a Monte Carlo method. The evolutionary dynamics (mutations and selection) are implemented assuming constant population size ($N = 200$ cells). Cells undergo fitness-dependent reproduction after every season which lasts $\tau_s$ Monte Carlo Steps of the CPM algorithm, and then the population is culled back to its original size. After this, environmental conditions are changed and a new season begins. Parameter values are motivated throughout this section, and summarised in *Table 1*. The custom software used for the simulations and to generate the figures is available at *Colizzi and Vroomans, 2020*.

### Cell dynamics

The model is a hybrid Cellular Potts Model implemented with the Tissue Simulation Toolkit (*Daub and Merks, 2015*). A population of $N$ cells exists on a regular square lattice $\Lambda_1 \subset \mathbb{Z}^2$. The chemotactic signal is located on a second plane $\Lambda_2$, of the same size and spacing as $\Lambda_1$. A cell $c$ consists of the set of (usually connected) lattice sites $\vec{x} \in \Lambda_1$ to which the same spin $s$ is assigned, that is, $c(s) = \{\vec{x} \in \Lambda_1 \mid \sigma(\vec{x}) = s\}$. The spin value is a non-negative integer, it is unique and positive for each cell, and it is used as the cell identifier. The medium is assigned spin $\sigma = 0$.

Cell movement arises from deformation of its boundaries through stochastic fluctuations. These fluctuations minimise a cell's energy, whose terms correspond to biophysically motivated cell properties (but see *Glazier, 2007* for a discussion on the statistical mechanics of the CPM). The energy minimisation occurs through the Metropolis algorithm (a Monte Carlo method), as follows. Fluctuations in cell boundary attempt to copy the spin value $\sigma(\vec{x})$ of a randomly chosen lattice site $\vec{x}$ to a site $\vec{x}'$ in its Moore neighbourhood. One Monte Carlo Step (MCS) consists of $L^2$ attempted copying events, with $L^2 = |\Lambda_1|$ (the size of the lattice, and $L$ one of its dimensions on a regular square lattice). Throughout this work $L = 500$. Whether an attempted spin copy is accepted depends on the contribution of several terms to the energy $H$ of the system, as well as other biases $Y$. A copy is always

accepted if energy is dissipated, that is, if $\Delta H + Y < 0$ (with $\Delta H = H_{\text{after copy}} - H_{\text{before copy}}$), and may be accepted if $\Delta H + Y \geq 0$ because of 'thermal' fluctuations following a Boltzmann distribution:

$$P(\Delta H, Y) = e^{\frac{-(\Delta H + Y)}{T}}$$

with $T = 16$ the Boltzmann temperature, a temperature-like parameter (in Arbitrary Units of Energy AUE) that controls the overall probability of energetically unfavourable fluctuations (allowing escape from local energy minima). The Hamiltonian $H$ of the system consists of two terms, corresponding to adhesion and cell size maintenance:

$$H = H_{\text{adhesion}} + H_{\text{cell size}}$$

The copy biases, or 'work terms', $Y$ consist of terms corresponding to cell migration and chemotaxis:

$$Y = Y_{\text{migration}} + Y_{\text{chemotaxis}}$$

## Cell adhesion

Adhesion between cells and to medium contribute to the Hamiltonian as:

$$H_{\text{adhesion}} = \sum_{(\vec{x},\vec{x}')} J(\sigma(\vec{x}), \sigma(\vec{x}'))(1 - \delta(\sigma(\vec{x}), \sigma(\vec{x}')))$$

where the sum is carried out over all the neighbour pairs $(\vec{x}, \vec{x}')$, and $\delta(\sigma(\vec{x}), \sigma(\vec{x}'))$ is the Kronecker delta which restricts the energy calculations to neighbouring lattice sites at the interface between two cells, or a cell and medium. $J(\sigma(\vec{x}), \sigma(\vec{x}'))$ is the contact energy between two adjacent lattice sites $\vec{x}$ and $\vec{x}'$ with different identity (i.e. $J = 0$ when $\sigma(\vec{x}) = \sigma(\vec{x}')$).

In order to calculate the values of $J(\sigma(\vec{x}), \sigma(\vec{x}'))$, we assume that cells express ligand and receptor proteins on their surface. Ligands and receptors are modelled as binary strings of fixed length $\nu$ (**Figure 1**, inspired by **Hogeweg, 2000**). Two cells adhere more strongly (experience lower $J$ values) when their receptors $R$ and ligands $I$ are more complementary, i.e. when the Hamming distance $D(R, I) = \sum_{i=1}^{\nu} \delta(R_i, I_i)$ between them is larger. Thus, given two cells with spin values $\sigma_1$ and $\sigma_2$ and their corresponding pairs of receptors and ligands $(R(\sigma_1), I(\sigma_1))$ and $(R(\sigma_2), I(\sigma_2))$:

$$J(\sigma_1, \sigma_2) = J_\alpha + 2\nu - D(R(\sigma_1), I(\sigma_2)) - D(R(\sigma_2), I(\sigma_1))$$

with $J_\alpha = 4$ chosen so that the final calculation yields values for $J(\sigma_1, \sigma_2)$ in the interval $[4, 52]$. For any particular receptor $R$ there is a single ligand $I$ which is maximally complementary, leading to a $J$ value of 4; and a single $I$ which is maximally similar, leading to a $J$ value of 52.

Adhesion of a cell with medium is assumed to depend only on the cell (the medium is inert, that is, $J(\sigma_{\text{medium}}, \sigma_{\text{medium}}) = 0$), and in particular it depends only on a subset of the ligand proteins of a cell. This subset consists of the substring of $I$ which begins at the initial position of $I$ and has length $\nu'$. The value of $J(\sigma_1, \sigma_{\text{medium}})$ is calculated as:

$$J(\sigma_1, \sigma_{\text{medium}}) = J'_\alpha + \sum_{i=1}^{\nu'} F(i)I_i$$

$$F(i) = \begin{cases} 4 & \text{if } i = 1 \\ 3 & \text{if } i = 2 \\ 2 & \text{if } i = 3 \\ 1 & \text{if } 4 \leq i \leq 6 \\ 0 & \text{if } i > 6 \end{cases}$$

with $J'_\alpha = 8$ and $F(i)$ a piece-wise defined function (a lookup table). The $J$ values range in the interval (**Du et al., 2015**; **Jacobeen et al., 2018**).

Encoding the energy values for cell adhesion in terms of receptor-ligand binding allows for flexibility and redundancy. Two cells that have the same receptors and ligands (i.e. given $R(\sigma_1), I(\sigma_1)$ and $R(\sigma_2), I(\sigma_2)$ with $R(\sigma_1) = R(\sigma_2)$ and $I(\sigma_1) = I(\sigma_2)$) can have any J value, by virtue of the particular receptor and ligand combination. The lookup table for the J value with the medium was chosen to

allow for a wide variety of possible J values with a small number of bits. Finally, implementing receptors and ligands in terms of binary strings allows for a simple evolutionary scheme, where mutations consist of random bit-flipping (more on this below). The numerical values of the various constants are chosen with four criteria in mind: (1) the receptor-ligand system has to be long enough that many different combinations are possible, so that its evolution is more open-ended; (2) two cells with random receptors and ligands do not - on average - adhere preferentially to each other or to the medium; (3) the range of adhesion energy must allow for strong clustering and strong dispersal while cells maintain their integrity; (4) although we are not fitting cell behaviour to any specific system, the adhesion energies must be in the typical range used to quantitatively model eukaryotic cells with CPM (*Graner and Glazier, 1992*; *Marée and Hogeweg, 2001*; *Ouchi et al., 2003*; *Magno et al., 2015*). With these constraints we set receptor and ligand lengths to $\nu = 24$. On average, two cells with random receptors and ligands will neither preferentially adhere to each other nor to the medium if their surface tension $\gamma = J(\sigma_{\text{cell}}, \sigma_{\text{medium}}) - J(\sigma_{\text{cell}}, \sigma_{\text{cell}})/2$ (see main text) is zero. We numerically checked (by generating a large number of ligands and receptors) that $\langle\gamma_{\text{(cells with random ligand receptors)}}\rangle = \langle J(\sigma_{\text{cell}}, \sigma_{\text{medium}}) - J(\sigma_{cell}, \sigma_{\text{cell}})/2\rangle = \langle[8, 20]\rangle - \langle[4, 52]\rangle/2 = 0$. Moreover $\gamma_{\max} = 18$ and $\gamma_{\min} = -18$ (parameter values in *Graner and Glazier, 1992*; *Marée and Hogeweg, 2001*; *Ouchi et al., 2003*; *Magno et al., 2015*).

## Cell size maintenance

Cell size $A(c) = |c(s)|$, the number of lattice sites that compose a cell, is assumed to remain close to a target size $A_T$ (equal for all cells). This is achieved by adding an energy constraint in the Hamiltonian that penalises cell sizes that are much larger or smaller than $A_T$:

$$H_{cellsize} = \sum_{c \in C} \lambda (A(c) - A_T)^2$$

with $C$ the set of cells $c$ present in the lattice configuration, and $\lambda$ a scaling factor for cell stiffness. This formulation captures the fact that cells are elastic objects that resist deformation from a preferred size ($A_T$). Unless otherwise specified, $A_T = 50$ lattice sites, chosen small enough to reduce computational load while large enough to avoid lattice anisotropy effects (*Magno et al., 2015*). The numerical value of $\lambda$ (set to 5 throughout the paper) is large enough to preserve cell size but not too large to freeze cells in place (see *Graner and Glazier, 1992*; *Ouchi et al., 2003* for details).

## Cell migration

We model migration (following *Beltman et al., 2007*) by biasing cell movement to their previous direction of motion $\vec{p}(c)$: extensions of a cell are energetically more favourable when they are closer to the direction of that cell's $\vec{p}$:

$$Y_{\text{migration}} = -\mu_{\text{p}} \cos(\theta_p)$$

Where $\mu_p$ is the maximum energy contribution given by migration, and $\theta_p$ is the angle between $\vec{p}$ and the vector that extends from the centre of mass of the cell to the lattice site into which copying is attempted. Every $\tau_p$ MCS the vector $\vec{p}$ is updated: its new value is the vector corresponding to the actual direction of displacement of the cell over the past $\tau_p$ MCS (scaled to unit) (*Figure 1*). Persistent migration occurs if $\tau_p \gg 1$, and captures the observation that a cell's cytoskeleton takes some time to re-polarise (*Ridley et al., 2003*). In line with previous CPM-based models of cell migration (*Vroomans et al., 2012*; *Vroomans et al., 2015*) we set $\tau_p = 50$ MCS. Note that all cells have the same $\tau_p$, but their initial moment of updating is randomised so that they do not update all at the same time.

## Chemotaxis

Individual cells are able to migrate towards the perceived direction of a chemoattractant gradient. The slope of the gradient is very shallow, making it difficult to perceive the direction over the typical length of a cell. Moreover, several sources of noise are introduced: cell's sampling error due to small size, noise due to integer approximation, and noise due to random absence of the signal.

The chemotactic signal is implemented as a collection of integer values on a second two dimensional lattice ($\Lambda_2 \subset \mathbb{Z}^2$, with the same dimensions as the CPM lattice). The (non-negative) value of a

lattice site represents the local amount of chemotactic gradient. This value remains constant for the duration of one season ($\tau_s$ MCS). The amount of chemotactic signal $\chi$ is largest at the peak, which is located at the centre of one of the lattice boundaries, and from there decays linearly in all directions, forming a gradient: $\chi(d) = 1 + (k_\chi/100)(L - d)$, where $k_\chi$ is a scaling constant, $d$ is the Euclidean distance of a lattice site from the peak of the gradient, and $L$ is the distance between the source of the gradient and the opposite lattice boundary; $L = \sqrt{|\Lambda_1|}$ for a square lattice. Non integer values of $\chi$ are changed to $\lceil \chi \rceil$ (the smallest integer larger than $\chi$) with probability equal to $\lceil \chi \rceil - \chi$, otherwise they are truncated to $\lfloor \chi \rfloor$ (the largest integer smaller than $\chi$). Moreover, the value of $\chi$ is set to zero with probability $p_{\chi=0}$ to create "holes' in the gradient. Setting $k_\chi = 1$ and $p_\chi = 0.1$ generates a shallow and noisy gradient. In a subset of simulations we used an alternative gradient, assumed to be generated by resources homogenously distributed on an entire side of the lattice, so that concentration isoclines are straight lines, see Appendix 2.4.

A cell has limited knowledge of the gradient, as it only perceives the chemotactic signal on the portion of $\Lambda_2$ corresponding to the cell's occupancy on $\Lambda_1$. We define the vector $\vec{\chi}(c)$ as the vector that spans from the cell's centre of mass to the centre of mass of the perceived gradient. Copies of lattice sites are favoured when they align with the direction of the vector $\vec{\chi}(c)$, i.e. when there is a small angle $\theta_c$ between $\vec{\chi}(c)$ and the vector that spans from the centre of mass of the cell to the lattice site into which copying is attempted (*Figure 1*):

$$Y_{chemotaxis} = -\mu_\chi \cos(\theta_c)$$

where $\mu_\chi$ is the maximal propensity to move along the perceived gradient, and is set to $\mu_\chi = 1$ in line with previous studies on cell migration (*Vroomans et al., 2012*) (chemotactic behaviour is robust to changes in $\mu_\chi$ however, see Appendix 1.6). A uniform random $\theta_c \in [0, 2\pi]$ is chosen whenever $|\vec{\chi}(c)| = 0$, that is, when, locally, there is no gradient (which may happen for very shallow gradients).

## Chemotaxis without gradient sensing

In a subset of simulations we implemented an alternative mechanism of collective chemotaxis (proposed by *Camley et al., 2016*) that does no rely on individual cells' gradient sensing. The mechanism works by combining three elements: cell–cell adhesion, contact inhibition of locomotion (*Mayor and Carmona-Fontaine, 2010*) and larger cell polarisation with higher concentration of the chemoattractant. The implementation of this mechanism in the CPM is straightforward. cell–cell adhesion was kept the same as explained above, and the chemoattractant is distributed to form the same gradient as in the previous paragraph. Every MCS each cell measures the average concentration of chemoattractant over the surface it covers $\chi(c)$ (note that this is a scalar). Then, in the copy biases $Y$ we substitute a new term $Y_{CIL}$ to the term $Y_{\text{chemotaxis}}$, with:

$$Y_{CIL} = \mu_{CIL} \chi(c)$$

and

$$\mu_{CIL} = \begin{cases} -3 & \text{if cell attempts spin copy into medium} \\ 0 & \text{if cell attempts spin copy with another cell} \\ 3 & \text{if medium attempts spin copy into cell} \end{cases}$$

This definition of $\mu_{CIL}$ introduces contact inhibition of locomotion by decreasing the probability that cells copy into each other, and increasing the probability that cells copy into medium.

## Evolutionary dynamics

A population of $N$ cells undergoes the cell dynamics described above for the duration of a season, i.e. $\tau_s$ MCS. At the end of the season the evolutionary dynamics take place. The evolutionary dynamics are decoupled from the cell dynamics for the sake of simplicity, and consist of fitness evaluation, cell replication with mutation, and cell death to enforce constant population size. The evolutionary experiments last 400 seasons - that is, 400 cycles of mutation-selection-dynamics. This value is larger than the time to reach evolutionary steady state in all simulations. Changes in $\tau_s$ result in qualitatively different evolutionary dynamics, as reported in the main text.

## Fitness evaluation

Fitness, that is, the probability of replication, is calculated at the end of each season for each cell. We do not include any explicit advantage or disadvantage due to multicellularity, and instead assume that fitness is based only on individual properties of the cells. Therefore, any multicellular behaviour is entirely emergent in this simulation.

The fitness $F(c)$ of a cell $c \in C$ depends only on the distance $d = d(c)$ of the centre of mass of a cell $c$ from the peak of the gradient as a sigmoid function which is maximal when $d = 0$, and decreases rapidly for larger values of $d$:

$$F(c) = \frac{1}{1 + \left(\frac{h_d}{d}\right)^2}$$

with $h_d$ being the distance at which $F(c) = 1/2$.

## Fitness cost of adhesion

In a subset of simulations (see Appendix 4.1) we include a fitness penalty due to the metabolic costs of maintaining adhesion with other cells. We compute the average amount of boundary a cell has in contact with other cells over the course of a season $\langle m \rangle$. The fitness of a cell $F(c)$ at the end of the season is then multiplied by a decreasing function of $\langle m \rangle$. For simplicity we use a linear function: $1 - c_m \langle m \rangle$, with $c_m$ the metabolic cost of adhesion, which can vary in $[0, 1]$. With small costs ($c_m \sim 0$) there is little penalty associated with adhering, whereas with large costs ($c_m \sim 1$) the fitness penalty punishes adhering cells more severely than non-adhering ones. When $c_m = 1$, a cell that spent the entire season completely surrounded by other cell has fitness 0, that is, it will not reproduce.

## Replication

For each cell $i \in C$ with fitness $F(i)$, the probability of replicating is $P(cell\ i\ replicates) = F(i) / \sum_{c \in C} F(c)$. We allow for $N$ replication events, each calculated with the same probabilities, choosing only cells that were already present in the previous season (so not their offspring). Cells with larger fitness may be chosen multiple times for replication.

Each replicating cell divides along its short axis (see e.g. *Hogeweg, 2000*), ensuring that related cells start close to each other at the beginning of the new season. One of the two daughter cells, chosen randomly, can re-enter the competition for replication. All the lattice sites belonging to the other daughter cell are assigned a new (unique) spin value and the cell can mutate its receptor and ligand. The bitstrings of the receptor and ligand may be mutated with a per-position probability $\mu_{R,I}$. Mutations flip individual bits (from 0 to 1, and vice versa).

Because repeatedly halving a cell's area would quickly lead to very small cells, we run a small number $\eta$ of steps of the cell dynamics (without cell migration and chemotaxis) between two replication events that affect the same cell, so that cells can grow back to target size ($\eta = 5$ MCS suffices).

## Death

After replication, there are 2$N$ cells on the lattice. In order to restore the initial population size $N$, half of the cells are removed from the lattice at random. When the initial population size is restored, the season ends. The new season begins by randomly placing the peak of a new gradient at the mid-way point of a randomly chosen boundary (different from the previous one). The remaining cells will then undergo the cell dynamics for the following $\tau_s$ MCS.

## Acknowledgements

We thank Paulien Hogeweg for constructive discussion. This research is supported by the Origins Center (NWA startimpuls). RM acknowledges his participation in the program 'Cooperation and the Evolution of Multicellularity' (2013) at the Kavli Institute for Theoretical Physics at the University of California, Santa Barbara that has inspired some of the ideas developed in this paper. All authors declare no conflict of interest.

## Additional information

### Funding

| Funder | Grant reference number | Author |
|---|---|---|
| Nederlandse Organisatie voor Wetenschappelijk Onderzoek | Nederlands Wetenschap Agenda StartImpuls | Enrico Sandro Colizzi Renske MA Vroomans |
| Nederlandse Organisatie voor Wetenschappelijk Onderzoek | NWO/ENW-VICI 865.17.004 | Roeland MH Merks |

The funders had no role in study design, data collection and interpretation, or the decision to submit the work for publication.

### Author contributions
Enrico Sandro Colizzi, Conceptualization, Software, Formal analysis, Validation, Investigation, Methodology, Writing - original draft, Project administration, Writing - review and editing; Renske MA Vroomans, Conceptualization, Software, Methodology, Writing - review and editing; Roeland MH Merks, Supervision, Methodology, Writing - review and editing

### Author ORCIDs
Enrico Sandro Colizzi ⓘ https://orcid.org/0000-0003-1709-4499
Renske MA Vroomans ⓘ http://orcid.org/0000-0002-1353-797X
Roeland MH Merks ⓘ https://orcid.org/0000-0002-6152-687X

### Decision letter and Author response
Decision letter https://doi.org/10.7554/eLife.56349.sa1
Author response https://doi.org/10.7554/eLife.56349.sa2

## Additional files
### Supplementary files
• Transparent reporting form

### Data availability
All data in this study are produced with custom-made software. The source code is freely available at https://github.com/escolizzi/Cell_Evolution_stickymoves (copy archived at https://archive.software-eheritage.org/swh:1:dir:911043b5acd69c35fb9b51e341dbb6dbd73dac52).

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

## Appendix 1

### Collective integration of spatial information drives the evolution of multicellularity

Collective chemotaxis

1.1 Diffusive exponent approximation

Main text *Figure 3d* shows the diffusion exponent for adhering and single cells in a cluster. Here we show how the figure is generated. In general, it is known that estimating single diffusion exponents from mean square displacement plots of anomalous diffusion is challenging (*Kepten et al., 2015*). Moreover, the diffusion exponent changes at different time interval. We therefore estimated the derivative of the log-log mean square displacement plot after interpolation with a polynomial, as shown in *Appendix 1—figure 1*.

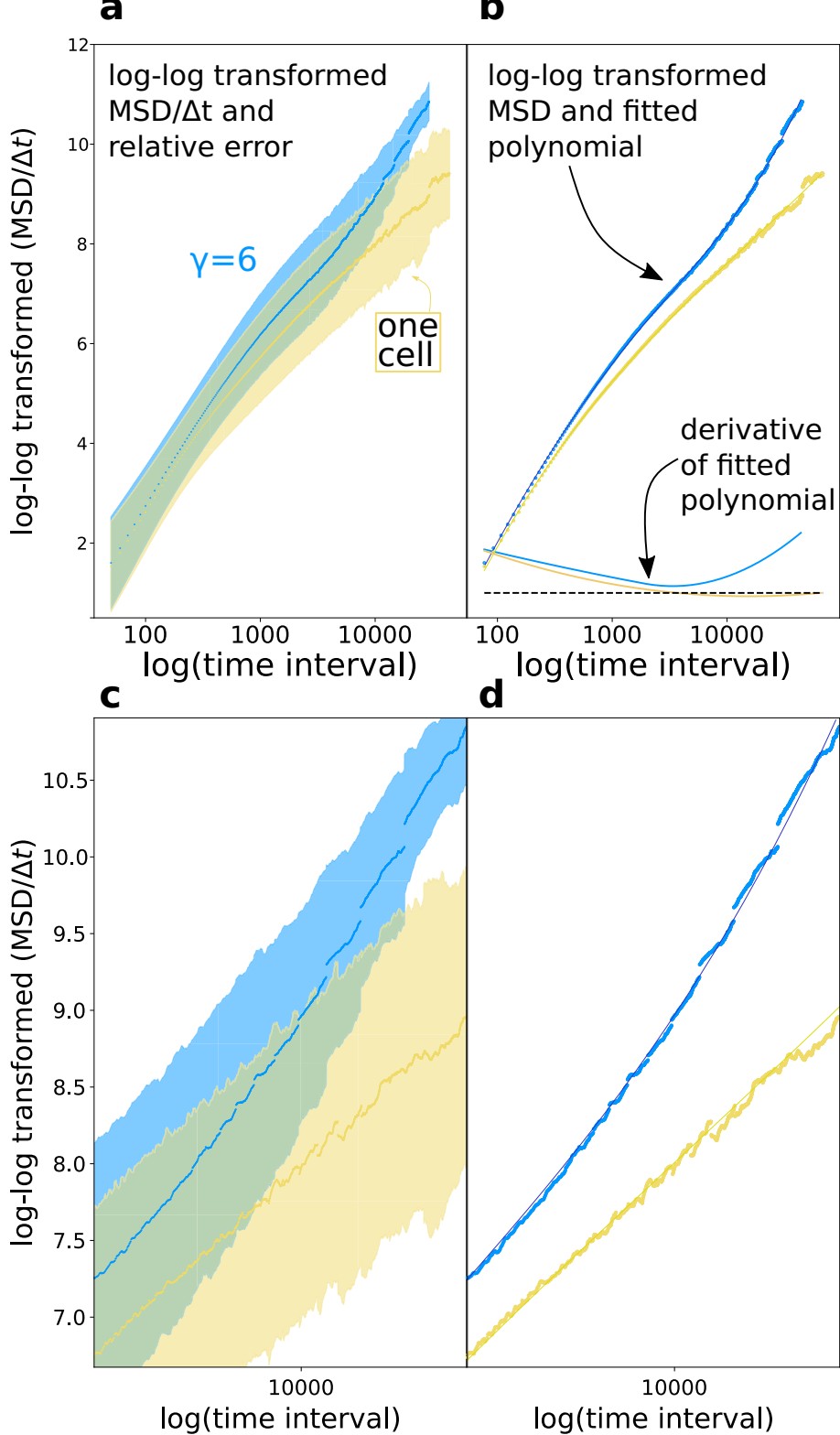

**Appendix 1—figure 1.** Diffusion exponent approximation. (**a**) We log-log transformed the data (the shaded area is the relative error $Var(MSD(\Delta t))/MSD(\Delta t)$ ). (**b**) We fitted a polynomial function to the data, then took the derivative of the polynomial function. (**c,d**) Magnifications of respectively (**a**) and (**b**).

## 1.2 Saturation of cluster speed with respect to cluster size

*Appendix 1—figure 2* shows that a sublinear and saturating increase of the average speed of a cluster of cells for larger number of cells in the cluster. The average speed of the cluster is obtained through linear fit of the displacement/time plot, where displacement is measured as the movement of the centre of mass of the cluster towards the peak of the gradient.

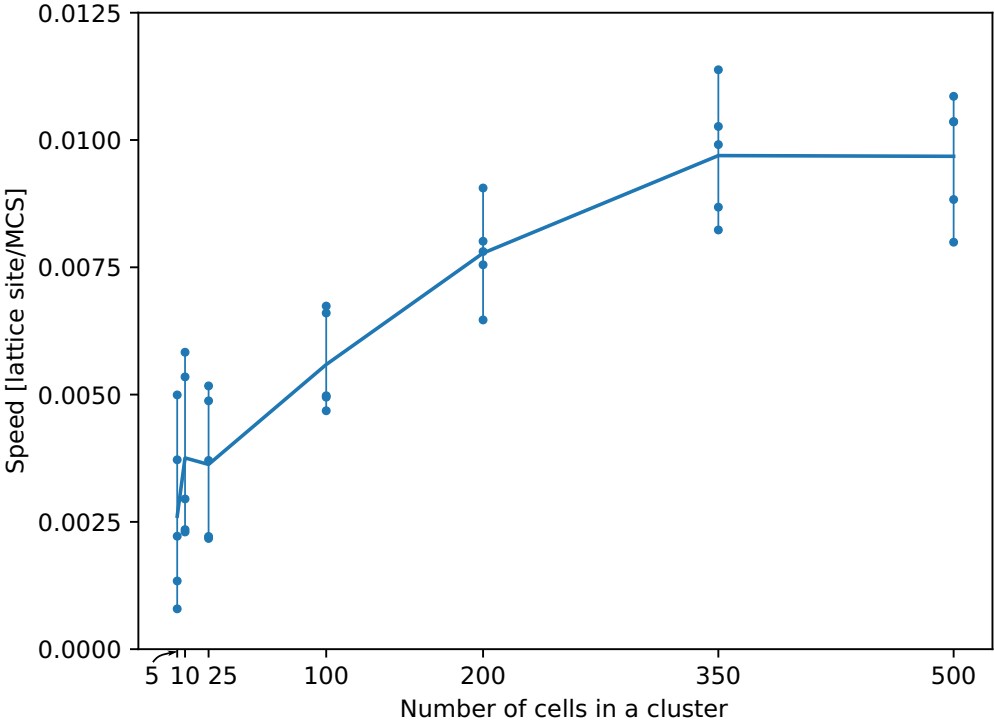

**Appendix 1—figure 2.** The speed of a cluster towards the peak of the gradient saturates with larger cluster sizes. For each cluster size, we ran five independent simulations. Each dot corresponds to one simulation. Their average (per cluster size) generates the line. All other parameters as in main text.

## 1.3 Indistinguishable relative movement of cells with and without a chemotactic gradient

Here we investigate whether cells in a cluster move differently when they are performing chemotaxis or not. *Appendix 1—figure 3* shows the flow field around moving cells in a cluster with or without a gradient, as devised by *Szabó et al., 2010*. In short, the flow field is calculated by taking each cell as a reference, and then rotating all other cells and their displacement vectors such that the reference cell displacement points to the right ($\vec{d} = \begin{bmatrix} x \\ 0 \end{bmatrix}$). Then the rotated displacement vectors are summed in bins at defined points in the neighbourhood (using all the cells as a reference, and using different time points) to obtain the average displacements in the neighbourhood (*Szabó et al., 2010*). In this case, the flow field shows that the relative movement of cells in a cluster is the same whether there is a gradient or not.

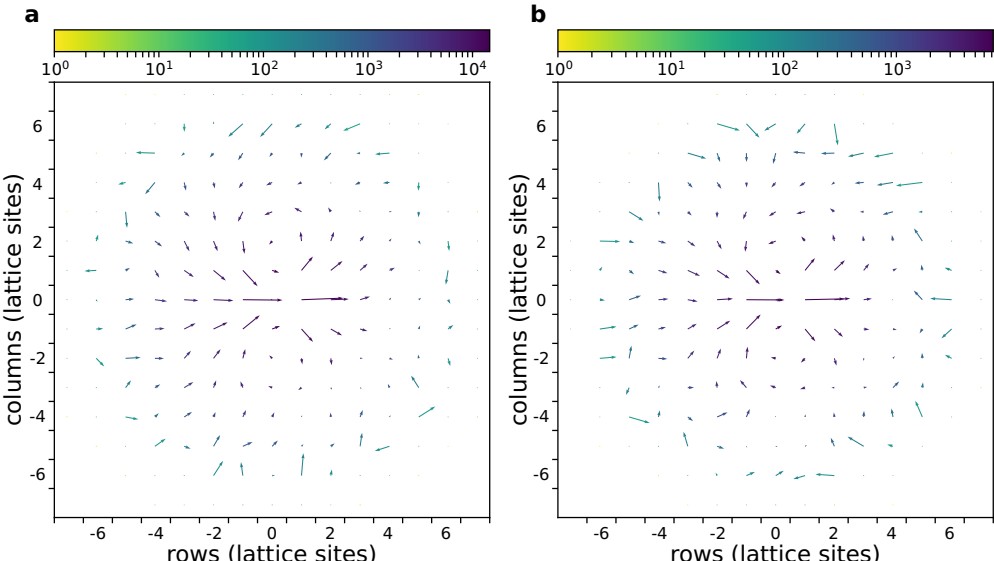

**Appendix 1—figure 3.** The flow field of a cluster of cells with and without gradient. (**a**) With chemoattractant gradient. (**b**) Without chemoattractant gradient. In both cases $N = 50$ cells with $\gamma = 6$ are placed at the centre of the field (All other parameters as in main text).

## 1.4 Chemotaxis with short persistence of migration and small persistence strength

*Appendix 1—figure 4* shows that chemotaxis occurs in a rigid cluster of strongly adhering cells, albeit at a slower speed than with default parameter (cf. main text *Figure 3a*). The lower persistence strength reduces the number of changes in the relative position of cells within the cluster.

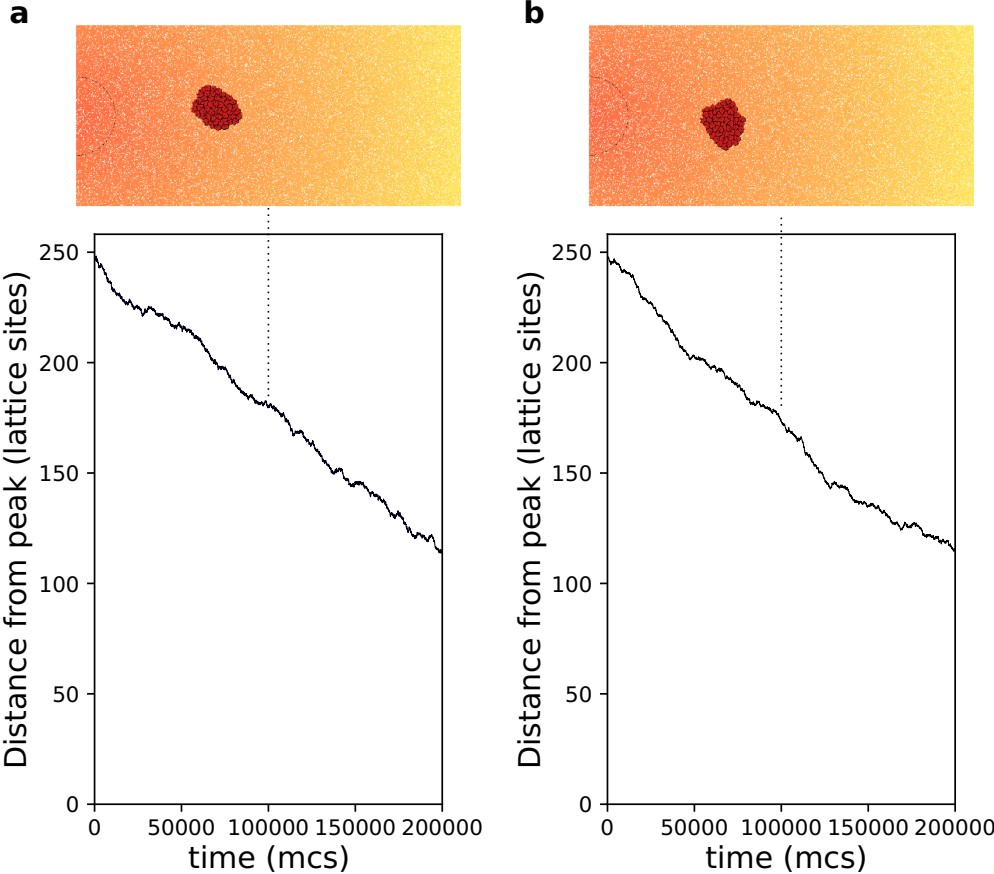

**Appendix 1—figure 4.** Chemotaxis of a rigid cluster. (**a**) $\tau_p = 5$. (**b**) $\mu_p = 0.5$. In both cases $N = 50$ cells with $\gamma = 6$ are placed on the right of the field and move towards higher concentration of the gradient (the semicircle indicates the resource location, where the gradient is highest. All other parameters as in main text).

## 1.5 Robustness of collective chemotaxis to changes in persistence strength

Persistent migration positively affects collective chemotaxis. *Appendix 1—figure 5* shows that this results (main text *Figure 3*) is robust to changes in the values of $\mu_p$, provided that $\mu_p$ is sufficiently larger than zero and not excessively large. When $\mu_p \sim 0$ persistent migration does not have a large effect on chemotaxis, and the chemotactic advantage of clusters is less pronounced. When $\mu_p > 5$ the model begins to break down, as the contribution of persistent migration to the energy function is too large compared to the other energies, and aberrant behaviour ensues (cells move forever in one direction regardless of adhesion or chemotaxis).

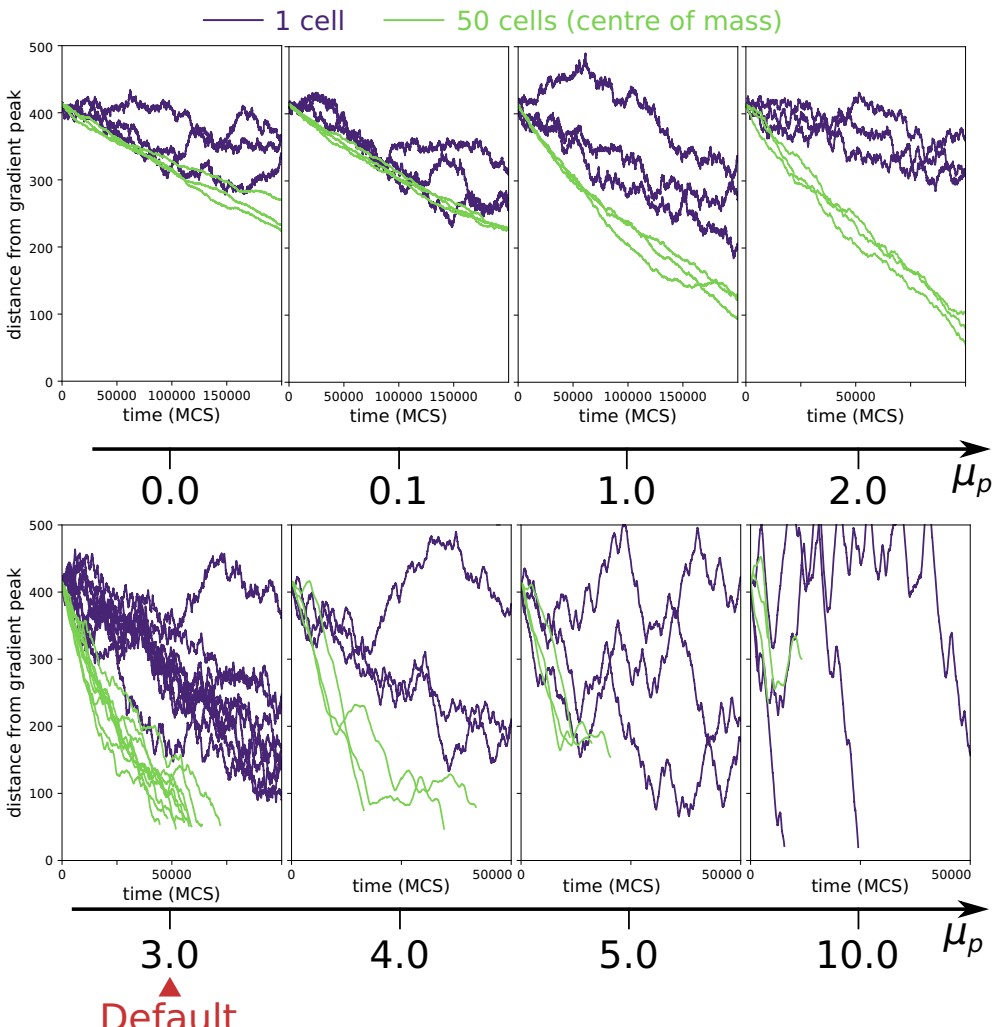

**Appendix 1—figure 5.** Collective *vs.* individual chemotaxis for different values of persistence strength $\mu_p \in [0, 10]$. The plots show the displacement over time of the centre of mass of a single cell (indigo) and that of a cluster of 50 cells (green). Note that the x axis is different in different plots. The value of $\mu_p$ used in main text is indicated by the Default sign. Three simulations are run for each parameter combination, except for the Default, where the same data of main text *Figure 3a* is shown. (All other parameters as in main text).

## 1.6 Robustness of collective chemotaxis to changes in chemotactic strength

Results presented in *Figure 3* are qualitatively unchanged (and quantitatively largely unaffected) by changes in chemotactic strength $\mu_\chi$, as shown in *Appendix 1—figure 6*.

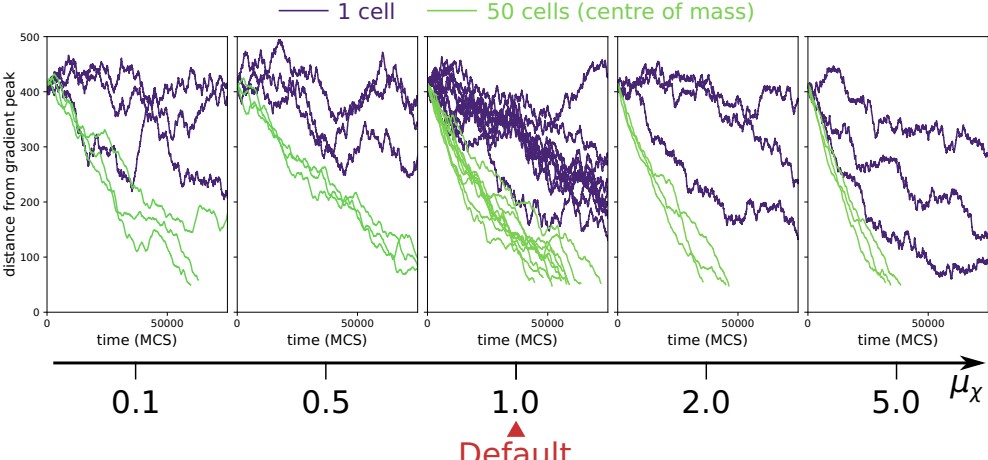

**Appendix 1—figure 6.** Collective *vs.* individual chemotaxis for different values of chemotactic strength $\mu_\chi \in [0.1, 5]$. The plots show the displacement over time of the centre of mass of a single cell (indigo) and that of a cluster of 50 cells (green). The value of $\mu_\chi$ used in main text is indicated by the Default sign. Three simulations are run for each parameter combination, except for the Default, where the same data of main text *Figure 3a* is shown. (All other parameters as in main text).

## 1.7 Chemotaxis of cells with different $A_T$

We explored the behaviour of different cell sizes and cell number by running simulations where the total area of the cells is kept constant, $NA_T = 5000$. We expect that large cells move with greater persistence towards the peak of the gradient than small cells, because they perceive a larger portion of the gradient, thus averaging out noise. Indeed, *Appendix 1—figure 7* shows that larger cells perform chemotaxis more efficiently than smaller cells, given the same chemotactic gradient.

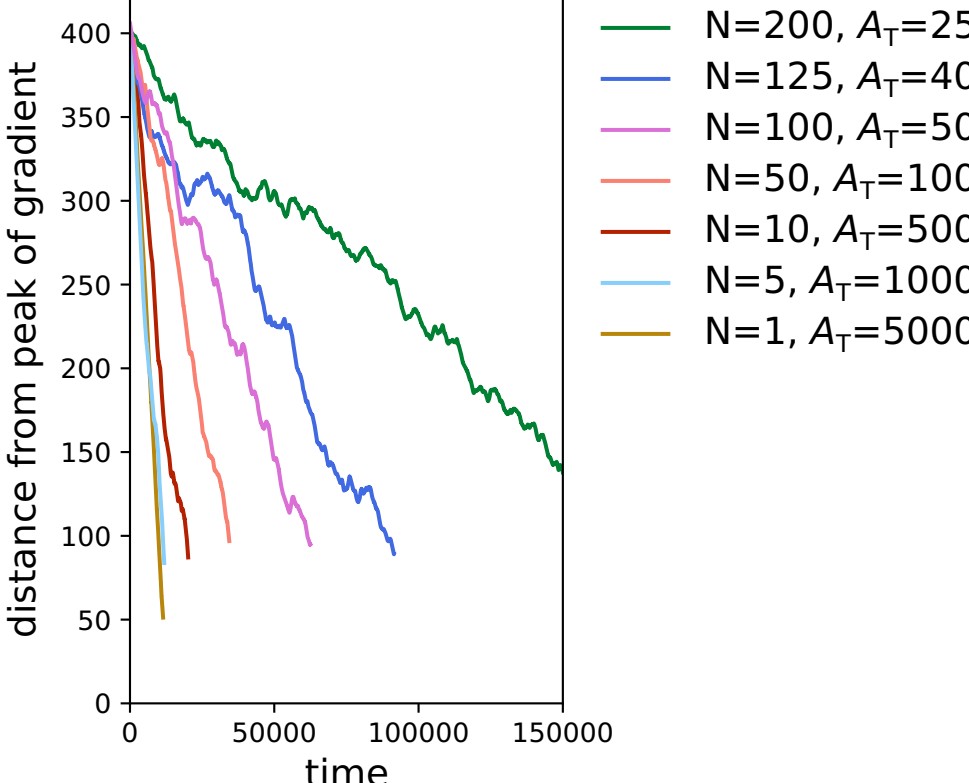

**Appendix 1—figure 7.** Large cells perform chemotaxis more efficiently than clusters of small cells.

Each line corresponds to one simulation with a given combination of number of cells $N$ and cell size $A_T$, and shows the distance of the centre of mass of the cluster of cells from the peak of the gradient over time. We kept the total volume of the cells constant in all simulations (i.e. $NA_T = 5000$). All other parameters (including the chemotactic signal) are the same as in main text.

## 1.8 Extraction of straight segments from cell tracks

For the contour plots in *Figure 3* of the main text, we extracted straight segments of the cells' trajectories, then measured the length of this segment and its angle with the direction of the source of the gradient. To identify these straight segments, we take increasingly longer intervals between the recorded cell positions, and measure how far the intermediate data points are positioned from the line spanning these two data points (*Appendix 1—figure 8A*). As soon as one of the data points has a distance greater than a threshold, we stop extending the interval and continue from the cell position at which the chosen segment ends (the threshold value is set to three lattice sites; this value is chosen because it is the largest integer smaller than the average cell radius, given a cell area = 50 lattice sites). In *Appendix 1—figure 8B*, the resulting segments are superimposed on cell position data from two simulations: one with a single cell and one with a cluster of adhering cells. While the overlap between the segment and the track itself varies, the length and orientation of the straight parts of the track are generally well-preserved in the segments.

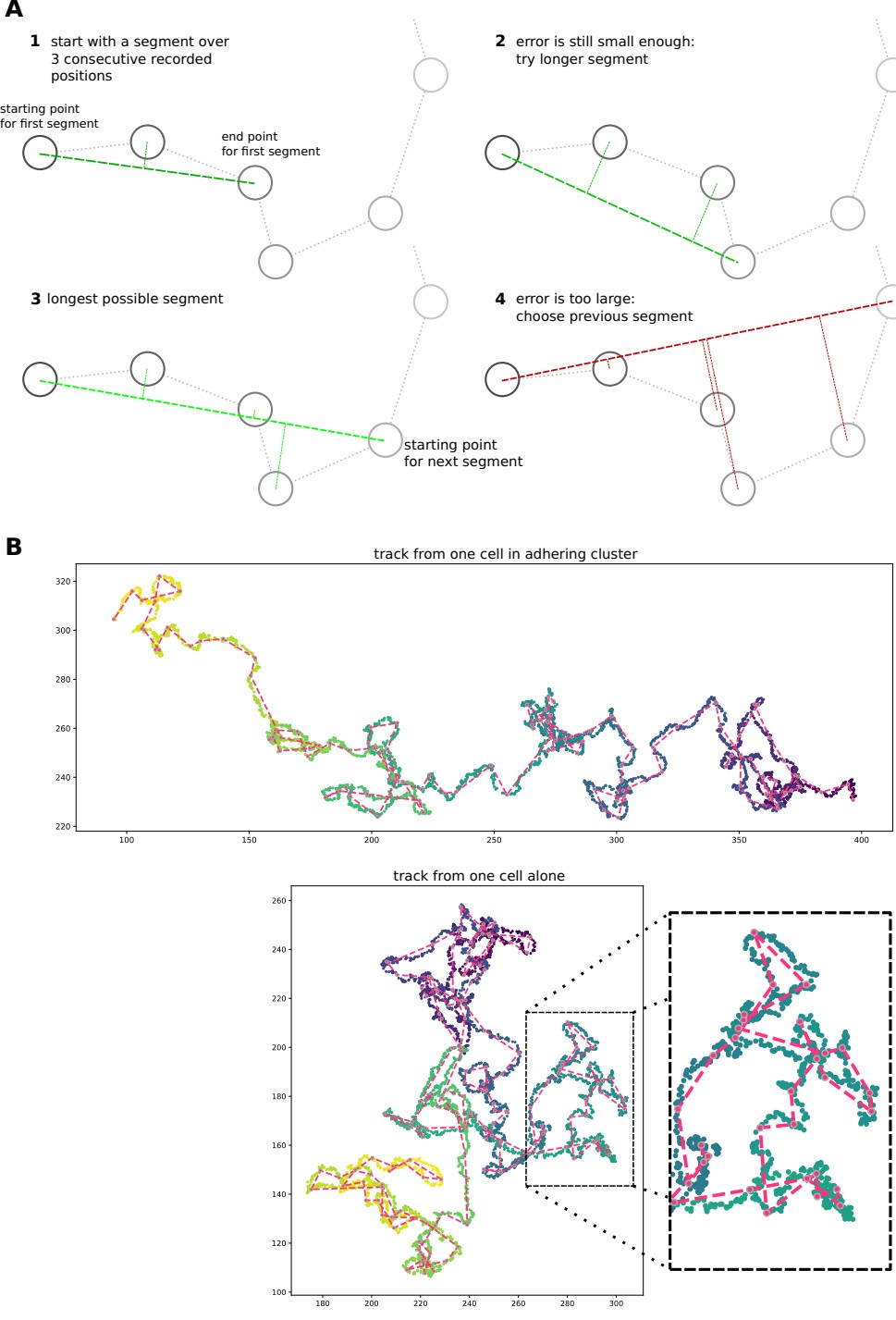

**Appendix 1—figure 8.** Simple algorithm for segment extraction. (**a**) Visual explanation of the algorithm, with a cartoon representation of a cell track with cell positions recorded at regular time intervals. Images 1-4 represent subsequent stages of the algorithm. For 1–3, the maximum distance of intermediate cell positions is still small enough, while for the segment in image 4 two intermediate positions are too far away. So the segment in image 3 will be used in the analysis, and we will start the algorithm from the fourth data point. (**b**) Two cell tracks from simulations, with the extracted segments superimposed in red.

## Appendix 2

### Evolution

### 2.1 Adhesion strength distribution for $\tau_s = \boldsymbol{100 \times 10^3}$ MCS

We collected ligands and receptors for all cells over 10 generations after evolutionary steady is reached, and we calculated for every cell the contact energy with the medium ($J(c_1, m)$) and with all other cells ($J(c_1, c_2)$) within the season in which it lived. A cell $c_1$ adheres to a second cell $c_2$ if their surface tension relative to medium (*Glazier and Graner, 1993*) is positive: $\gamma_{c_1,c_2} = J(c_1, m) - J(c_1, c_2)/2 > 0$. We compared the resulting distribution with the distribution of $\gamma$ values obtained by generating $10^5$ pairs of ligand and receptors, and calculating $\gamma$ from them. *Appendix 2—figure 1* (blue) shows that random cells, on average, are neither adhering nor non-adhering (the calculated $\langle \gamma \rangle$ was 0.00). *Appendix 2—figure 1* (black) shows the distribution of all vs. all cells adhesion energy. Cells adhere to one another strongly ($\langle \gamma \rangle = 4.85$), and with little variability when compared to the random ligands and receptors (blue): the calculated variance is respectively (4.95 and 10.95) This method however neglects that cells self-organise within a season to minimise contact energy. We therefore extracted the adhesion energies of cells that are in contact with one another during a season and repeated the procedure outlined above. *Appendix 2—figure 1* (red) shows that the surface tension distribution for observed contacts is on average higher and more peaked than the all vs. all distribution ($\langle \gamma \rangle = 5.82$ and variance 3.54).

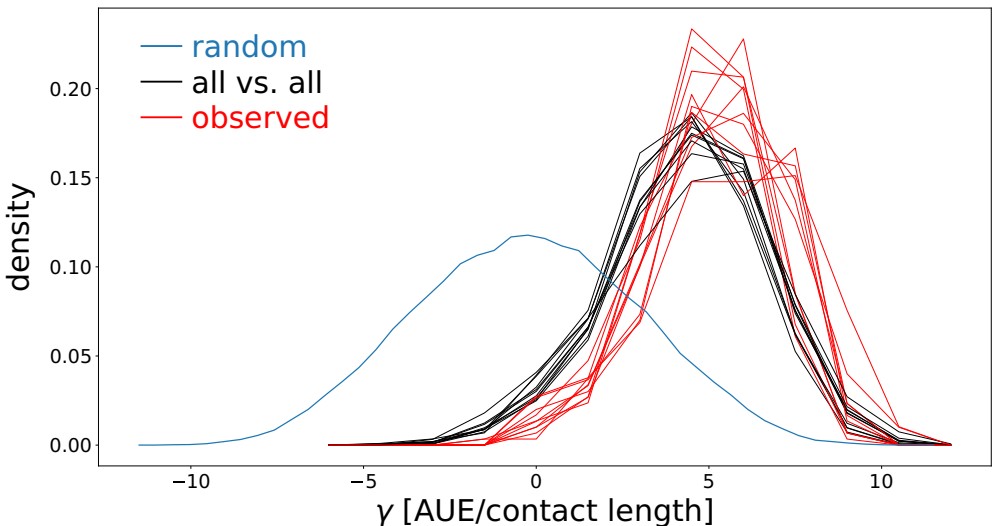

**Appendix 2—figure 1.** Surface tension distribution for a population of cells that evolve adhesion, compared to the distribution of adhesion strength for randomly generated ligands and receptors. The data for adhering cells are taken from the same simulation used for main text *Figure 5a*, over 10 seasons after reaching evolutionary steady state with $\tau_s = 100^3$ MCS. Black: all vs. all surface tension distribution (all possible pairwise cell interaction energies tested); red: observed distribution. All parameters are as in main text *Figure 5a*. Blue: surface tension of random ligand receptors ($10^5$ pairs were generated). AUE: Arbitrary Units of Energy (see *Table 1*).

### 2.2 Evolution of multicellularity with alternative values of persistent migration strength

We tested the robustness of the evolution of multicellularity when the value of persistent migration strength was changed in the interval $\mu_p \in [1, 5]$. Because the evolution of multicellularity relies on collective chemotaxis, which in turn is sped up by larger values of $\mu_p$, we expect that multicellularity evolves at smaller $\tau_s$ when $\mu_p$ is larger. *Appendix 2—figure 2* shows that this is indeed the case. Moreover, it shows that $\mu_p = 1$, where collective chemotaxis is slow, multicellularity evolves only if $\tau_s > 250 \times 10^3$ MCS (we tested $\tau_s = 500 \times 10^3$ MCS, which is at the limit of computational feasibility).

For $\mu_p = 5$ collective chemotaxis is so fast that no uni-cellular strategy was found (the smallest value tested was $\tau_s = 1 \times 10^3$ MCS).

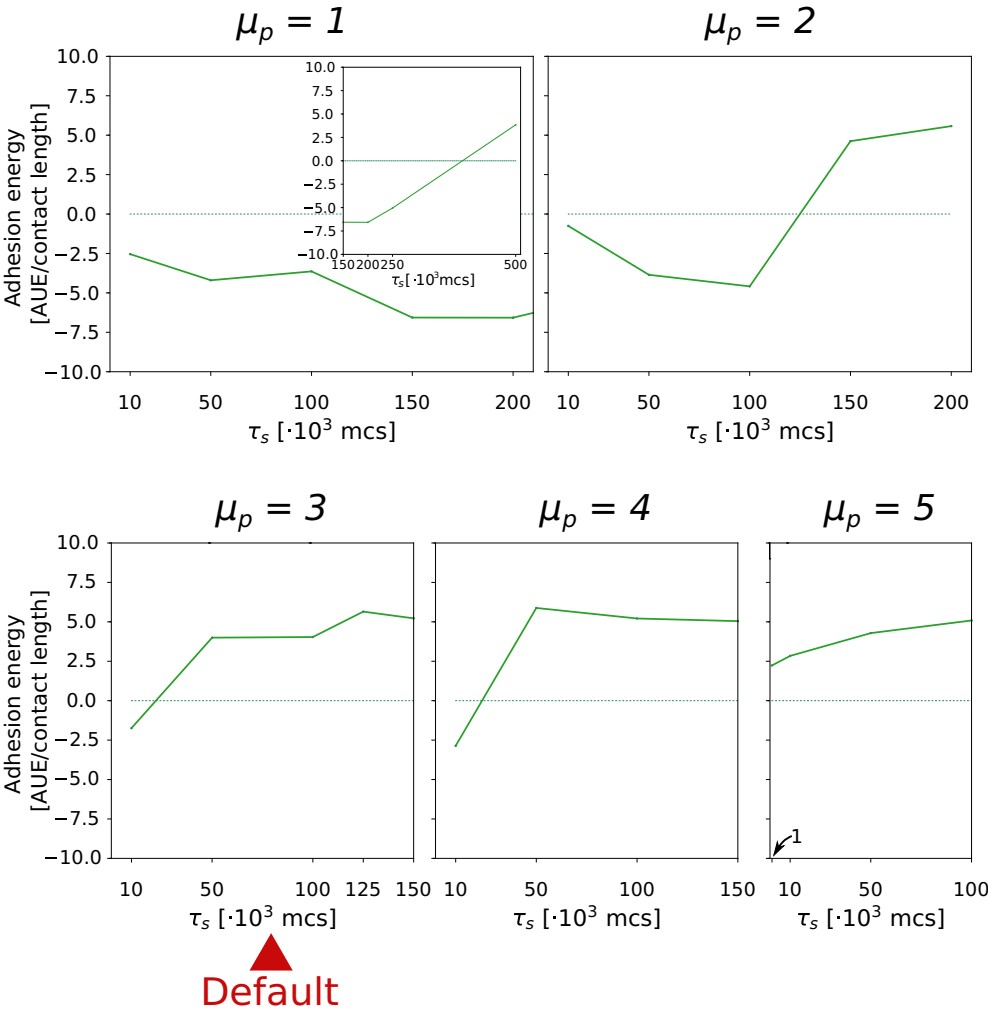

**Appendix 2—figure 2.** The evolution of multicellularity (and uni-cellularity) for different values of persistent migration strength $\mu_p \in [1, 5]$. For $\mu_p = 1$, the inset shows the surface tension for $\tau_s > 200 \times 10^3$ MCS. All other parameters and initialisation are as in main text *Figure 5a*.

## 2.3 Evolution of multicellularity with alternative values of persistent migration strength

. We tested the robustness of the evolution of multicellularity when the value of chemotactic strength was changed in the interval $\mu_\chi \in [0.5, 2]$. In general, results were robust in this interval: multicellularity evolves during sufficiently long seasons ($\tau_s$), and otherwise the uni-cellular strategy evolves (*Appendix 2—figure 3*). We found that the transition to multicellularity happens at longer seasons for larger values of $\mu_\chi$, despite the cluster moving faster (see *Appendix 1—figure 6*). We hypothesise that this is due to increased chemotaxis efficiency of individual cells at larger $\mu_\chi$.

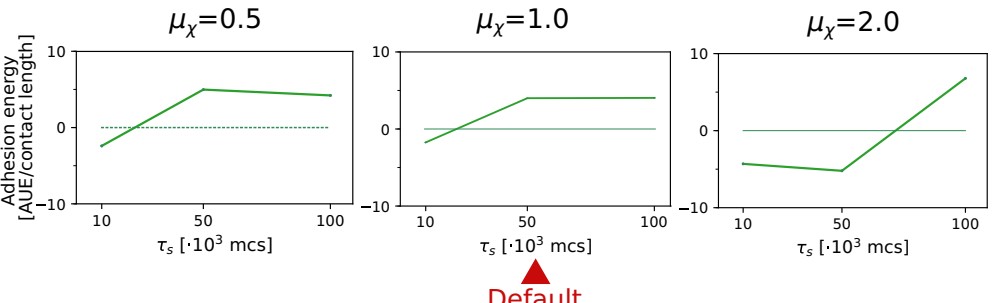

**Appendix 2—figure 3.** The evolution of multicellularity (and uni-cellularity) for different values of chemotactic strength $\mu_\chi \in [0.5, 2]$. All other parameters and initialisation are as in main text *Figure 5a*.

## 2.4 Evolution of multicellularity with a 1D gradient

Throughout the main text, the chemoattractant gradient decreases radially from the point where resources are assumed to be located. As cells migrate closer to the peak of the gradient, chemoattractant concentration isoclines are more curved, meaning that the direction of the gradient is progressively more precise, and thus chemotaxis may speed up. We tested if results are robust to modifying the gradient so that isoclines are straight and parallel. We ran a series of evolutionary simulations assuming that resources are distributed on the entire side of the lattice (instead of only on one point), so that chemoattractant concentration isoclines are straight lines, and we let the side change randomly every season. The numerical value of chemoattractant concentration at any point is given by the same function as for the radial gradient (see Materials and methods) $\chi(d) = 1 + (k_\chi/100)(L - d)$, where $d$ is here the distance of the point from the side of the lattice where the concentration is largest. Fitness is then evaluated based on the distance from the side of the lattice, with the same fitness function as in Materials and methods ($F(d)$, where $d$ is the distance of the centre of mass of the cell from the side of the lattice where the concentration is largest).

Results were largely unaffected by this change (*Appendix 2—figure 4*). Multicellularity evolves for $\tau_s > 50$, which is only a slightly longer seasons duration than in main text *Figure 5*.

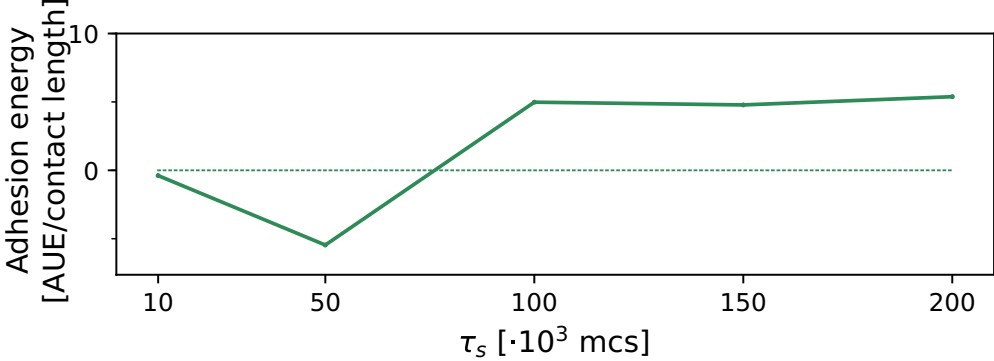

**Appendix 2—figure 4.** The evolution of multicellularity (and uni-cellularity) when resources are spread for a chemoattractant gradient that decreases parallel from resources distributed on the entire side of the lattice. All other parameters and initialisation are as in main text *Figure 5*.

## 2.5 Evolution of multicellularity with a steep noiseless gradient

In the main text, the gradient is chosen such that individual cells perform chemotaxis inefficiently, as this provides a selectable incentive to aggregate when seasons are sufficiently long. When seasons are short, a uni-cellular strategy evolves where cells undergo fast dispersal after replication. Here we test the robustness of these results when the gradient is very steep, that is, $k_\chi = 10$, and noiseless, $p_{\chi=0} = 0$. We found that multicellularity evolves at much shorter seasons ($\tau_s \geq 5 \times 10^3$ MCS) than in main text (*Appendix 2—figure 5*). This is because collective chemotaxis is faster than individual

chemotaxis, and because the uni-cellular strategy is slower than chemotaxis when the gradient information is precise.

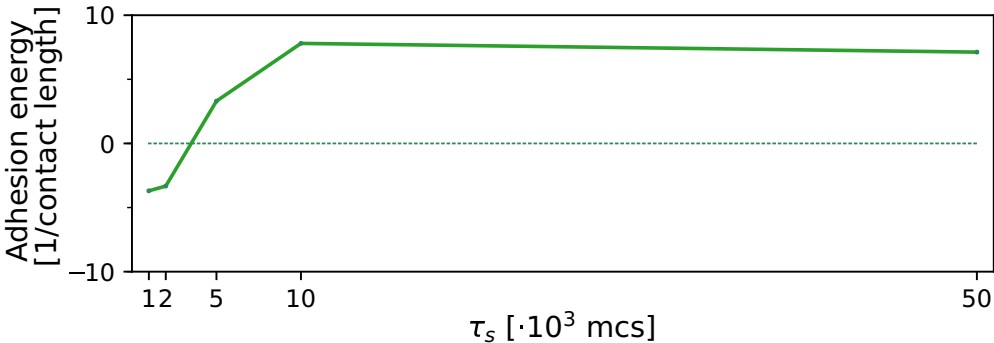

**Appendix 2—figure 5.** The evolution of multicellularity (and uni-cellularity) with a steep, noiseless gradient ($k_\chi = 10$, $p_{\chi=0} = 0$). All other parameters and initialisation are as in main text *Figure 5*.

## Appendix 3

### Alternative chemotaxis mechanism

### 3.1 Evolution of multicellularity when a cell cluster - but not individual cells - can perform chemotaxis

In the main text, we showed that the emergent organisation of a cluster of adhering cells increases the efficiency of chemotaxis and can be exploited as an indirect selection pressure for evolving adhesion when seasons are sufficiently long. Main text results are based on the assumption that individual cells are able to perform chemotaxis, albeit inefficiently. Here show that relaxing this assumption does not change results. To this end, we implemented a model developed by *Camley et al., 2016* where collective chemotaxis emerges from the combination of contact inhibition of locomotion and stronger cell membrane polarisation when the chemoattractant concentration is larger. Because individual cells can only sense the concentration of the signal (and does not perceive the direction of the gradient), chemotaxis can only emerge as a property of a cell cluster. We modified the energy function by replacing the chemotaxis term with a new term that favours extension of cells into the medium in a concentration dependent manner and disfavours extensions towards other cells (with strength $\mu_{CIL}$, see Materials and methods Section).

In *Appendix 3—figure 1a* we show that individual cells do not sense the gradient and therefore perform random walks, while adhering clusters move efficiently up the gradient through emergent chemotaxis. In *Appendix 3—figure 1b* we show that this alternative form of collective behaviour leads to similar results as in main text: multicellularity evolves for sufficiently large values of season duration $\tau_s$, and a uni-cellular strategy based on dispersal evolves for shorter season duration.

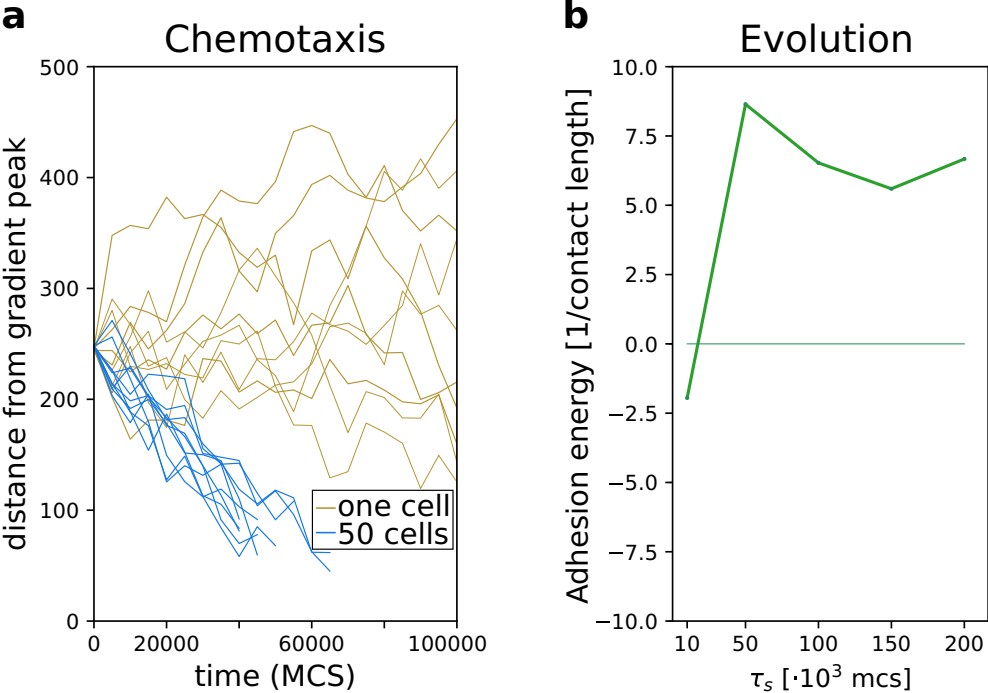

**Appendix 3—figure 1.** Emergence of collective behaviour and evolution of multicellularity are robust to changing the mechanism of chemotaxis. (**a**) The emergence of collective chemotaxis when individual cells cannot sense the gradient; (**b**) the evolution of multicellularity (and uni-cellularity) under these conditions. $\mu_{CIL}$ - the strength of contact inhibition of locomotion is defined in the Materials and methods section. All other parameters and initialisation are as in main text *Figure 5*.

## Appendix 4

### Evolution with cost

### 4.1 The effect of costly adhesion on the evolution of multicellularity

In the main text, we showed that the indirect benefit of collective chemotaxis is sufficient to select for multicellularity when seasons are sufficiently long, and otherwise a uni-cellular strategy evolves. Here we explicitly incorporate costs for multicellularity. We assume that a metabolic cost is paid to maintain adhesion between cells. In one MCS, a cell incurs a cost proportional to the fraction of its boundary that is in contact with other cells $c_m \langle m \rangle$. This cost is summed for the duration of a season, so that cells that spend a long time in contact with other cells incur a larger metabolic cost. At the end of the season, a fitness penalty proportional to the overall (per-season) cost of adhesion is applied to each cell. For $c_m = 0$ this model reduces to the main text model, i.e. there are no costs for adhering with other cells. The maximum cost of adhesion is scaled to $c_m = 1$; in this case, a cell that is completely surrounded by other cells for the entire duration of a season (i.e. $\langle m \rangle = 1$) will not reproduce at all - effectively making dispersal the only viable strategy. Overall, this formulation of costs decreases fitness for cells that spend more time in contact with others. See main text Materials and methods for details.

We studied the evolution of multicellularity with costly adhesion for several value of the metabolic cost $c_m \in [0, 0.75]$, over a range of season duration $\tau_s \in [10, 100]^3$ MCS. Results are shown in *Appendix 4—figure 1*. Results are identical to main text *Figure 5* when costs are small ($c_m < 0.01$): multicellularity evolves when $\tau_s \leq 50 \times 10^3$ MCS. With larger costs $0.1 \leq c_m \leq 0.5$, the evolution of multicellularity is shifted to larger season duration. Only when costs are sufficiently large ($c_m = 0.75$), multicellularity did not evolve over the values of $\tau_s$ that we tested. Altogether, the evolution of multicellularity for longer season duration is a robust outcome of the model, provided that the cost of adhesion is not too high.

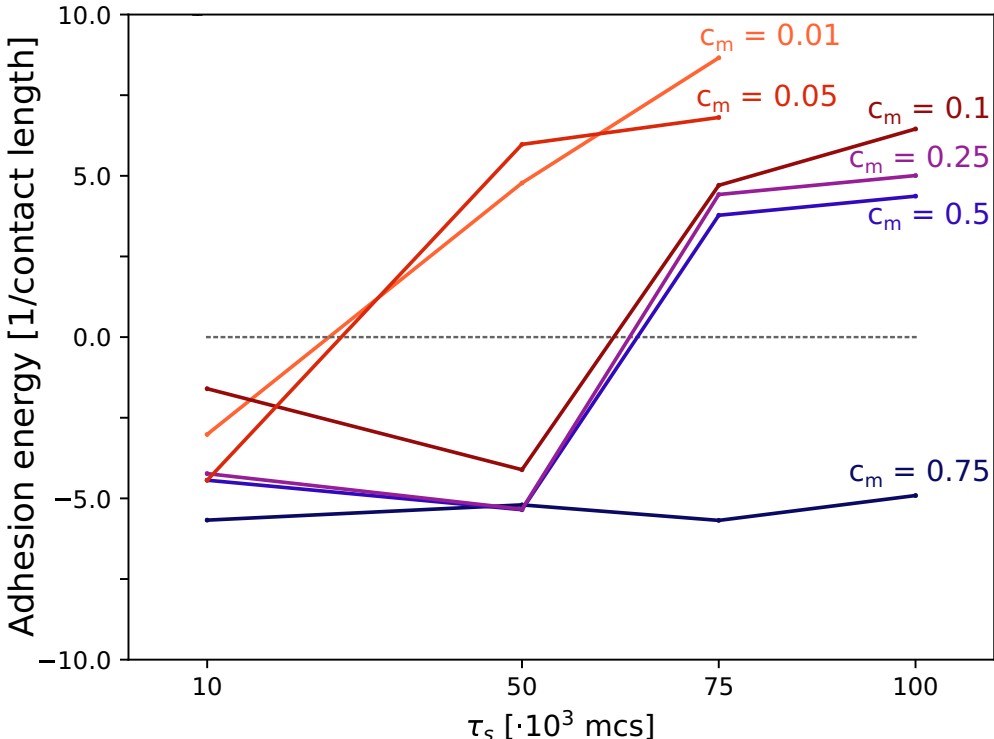

**Appendix 4—figure 1.** The evolution of multicellularity (and uni-cellularity) when adhesion is costly. Different lines correspond to the evolutionary steady state at different season duration $\tau_s$ for different values of costs $c_m$, as indicated in the figure. All other parameters and initialisation are as in main text *Figure 5*.

