## [Decision Letter]

**Acceptance summary:**

This work is a computational work on the evolution of multicellularity. Inspired by the life cycle of *Dictyostelium*, the authors develop a mathematical model that incorporates cellular aggregation and chemotaxis in a cyclic environment and show how the greater fidelity of multicellular chemotaxis leads to selection for that state. The model is an enlightening case study of this very important evolutionary issue.

**Decision letter after peer review:**

Thank you for submitting your article "Evolution of multicellularity by collective integration of spatial information" for consideration by *eLife*. Your article has been reviewed by three peer reviewers, and the evaluation has been overseen by a Reviewing Editor and Aleksandra Walczak as the Senior Editor. The reviewers have opted to remain anonymous.

The reviewers have discussed the reviews with one another and the Reviewing Editor has drafted this decision to help you prepare a revised submission.

Summary:

In this paper, the authors consider the problem of evolutionary transitions to multicellularity, and in particular the case in which aggregation drives the process. Inspired by the life cycle of *Dictyostelium*, they consider a model in which cells (moving on a grid) search for resources and can adhere to each other based on the match between ligand and receptors on their surfaces. All of this takes place in the context of a chemotactic march towards a local chemoattractant within one temporal "season", after which fitness-dependent reproduction occurs, the population is culled back to its starting size, and the environmental conditions are changed.

The reviewers all are of the opinion that this work provides an interesting perspective on a possible mechanistic basis of 'collective-level' function, that stems from physical interactions among cells in the absence of explicitly modelled costs and benefits of single cell's choices. At the same time, the reviewers were clear that there are many aspects of the model and the modelling approach that are not clear, unnecessarily complicated or not well justified. In light of these, major revisions to the paper will be necessary, as explained below.

Essential revisions:

1) Considering the paper as a whole, there are far too many things happening at once to draw any meaningful conclusions. There is the complexity of adhesion, the nature of the chemotaxis, the temporal switching between seasons, and the reproduction process. Each of these is explored to a limited extent, and it is unclear which are absolutely crucial to the conclusions reached and how sensitive the conclusions are to the assumptions made.

2) Regarding the definition of the model itself, the reviewers find it inappropriate to relegate so much of that explanation to the Materials and methods section. The very large number of parameters (18) in Table 1 needs to be made clear (and that table should be referenced – it does not appear to be at present). Please explain more of the model in the body of the paper.

3) The reviewers are supportive of abstract models, but inasmuch as the authors have set up a physical/biological scenario with familiar processes (chemotaxis, adhesion) it would have been very helpful to have justified the kinds of dimensionless parameters that characterize the model in terms of real physical and biological features.

4) The essence of a Monte Carlo simulation is the definition of an energy function and a temperature, which together yield a Boltzmann factor that is used to decide if an attempted step is taken. The authors do not make clear in the main body of the text that they are performing a Monte Carlo calculation (that is only specified in the Materials and methods section). They refer to MCS (Monte Carlo Steps) in the body of the paper without defining that term. But the larger question is why this kind of nonequilibrium biological system should have such an energy, and what would be the biological significance of the temperature? In addition, of course, the "steps" taken are those of Monte Carlo algorithm and have no direct interpretation in terms of real time.

5) The presentation of the model and the main results lack clarity in some key aspects:

a) the relation between cell–cell and cell-medium adhesion and surface tension (subsection “Strongly adhering cells perform efficient collective chemotaxis We first assessed how well groups of cells with different adhesion strengths”) is not explained, so it is not really clear what negative surface tension means.

b) as surface tension pools two different kinds of adhesion, does it mean that in a certain sense adhesion to the surface can be traded off against adhesion between cells? This is important to know in connection to experiments.

c) since the measure of sequence complementarity is symmetric, why does one need to suppose the existence of both a ligand and a receptor? Would it change anything if cells were characterized by only one sequence? If yes, it would be interesting to know if at the end of the numerical experiment ligand and receptor evolve to be the same or if 'molecular' diversity is maintained.

d) the process of cell division/regrowth and the fact that cells do not change position from one season to the next should be more clearly explained in the main text.

e) what is the initial spatial distribution of cells at the beginning of every season, and if this matters (many models assume aggregation-dispersal cycles, that does not seem to be the case here), should be specified or repeated in the evolutionary section.

f) Figure 5 should depict a case of bistability: now it is not clear that different evolutionary outcomes are associated with differences in the initial surface tension, rather than in the initial cell configuration. It would by the way be interesting to see if the second also gives rise to bistability.

6) Cell migration (subsection “Cell migration”) is defined in terms of the actual direction of the cell over the past steps. This seems to build in persistence, and would appear to have a profound effect on the dynamics. Is this the case?

7) In general, it would be useful if statements like "In our case, aggregation leads to a highly efficient search strategy, guided by long-range, albeit noisy, gradients." (Discussion section) could be made more quantitative. For instance, one would like to get a sense of whether the conclusions are robust to changes in (at least a few important) parameters. One would expect so from results in active matter physics, but it would be useful of the authors could argument it and indicate when they expect different conclusions to hold. Moreover, what is the role of the particular gradient chosen here in 'focalizing' the formation of multicellular groups (would an essentially 1-D gradient, where isolines are parallel, do the job?) and of its intensity/spatial variation (in the movie, one sees that the centre of the gradient changes among four positions, does it matter?).

8) The authors claim that, in contrast to previous work, the increased fitness of the aggregates (better ability to perform chemotaxis) is an emergent property. The reviewers struggled to find a physical/mathematical explanation as to why such a relationship exists in the model but it appears that subsection “Chemotaxis” contains the mechanism. The text speaks of the "centre of mass of the perceived gradient". Unless we are mistaken, such a quantity averages over the individual constituent's contributions in such a way that larger cells will have more accurate measurements of the gradient. This is just the law of large numbers. If this is the case, then this feature is not an emergent property at all, but is part of the definition of the model. Please clarify. If the above critique is correct, then why bother with the complex model? The authors could just use the fact that larger aggregates are better at chemotaxis for the reason given and proceed from there.

The above suggests that the authors have basically put the answer in from the beginning. The model has the explicit feature that those that perform chemotaxis better reproduce more. So of course, that will be reinforced. But multicellularity has costs and benefits, and the model does not appear to contain any costs associated with multicellularity. In real biological examples there are many – the increased metabolic cost of the structures that hold cells together, greater need for regulatory genetic networks, etc.

9) The referencing of the text to Figure 3 is all mixed up, leaving both text and figure hard to follow. -The authors should revise this section and make sure that they clearly state if higher chemotactic performance arises due to longer persistence of cell clusters only or due to longer persistence and higher chemotactic accuracy of whole cell cluster. Varennes et al., (2017) and manuscripts citing this work give measures for chemotactic accuracy within cell populations. – Figure 3D should show error bars. Annotation of Figure 3F should be detailed, what is bar{X}? Is this the local gradient including noise or averaged on which scale.

10) The assessment time scale emerges as a decisive factor – it appears as a theoretical construct right now. What could it correspond to in the real world? Please discuss.

11) As for the particular details of the model, it is left unsaid in the main text but stated in the Materials and methods section that there is a preferred cell size A_T and a harmonic energy around that size. As the target size is (Table 1) some 50 pixels, we are confused, as it seems that each "cell" occupies one lattice size. This energy would then clearly bias the system to aggregate already. Please clarify. The use of the term "pixel" for a lattice site is confusing.

12) The literature overview appears limited – please revise and consider recent work for example but not limited to Varennes et al., (2017); Jacobeen et al., (2018). The authors should also discuss Guttal and Couzin, 2010. And they should acknowledge relevant literature exploring, for example, similar issues in the Volvocales; Solari er al., (2006); Solari, Kessler and Goldstein, (2013).

[Editors' note: further revisions were suggested prior to acceptance, as described below.]

Thank you for submitting your article "Evolution of multicellularity by collective integration of spatial information" for consideration by *eLife*. We regret that delay in reaching a decision on your revised manuscript, due in part to challenges presented by the pandemic. Your article has been reviewed by three peer reviewers, and the evaluation has been overseen by a Reviewing Editor and Aleksandra Walczak as the Senior Editor. The reviewers have opted to remain anonymous.

The reviewers have discussed the reviews with one another and the Reviewing Editor has drafted this decision to help you prepare a revised submission.

Summary:

In this paper, the authors consider the problem of evolutionary transitions to multicellularity, and in particular the case in which aggregation drives the process. Inspired by the life cycle of *Dictyostelium*, they consider a model in which cells (moving on a grid) search for resources and can adhere to each other based on the match between ligand and receptors on their surfaces. All of this takes place in the context of a chemotactic march towards a local chemoattractant within one temporal "season", after which fitness-dependent reproduction occurs, the population is culled back to its starting size, and the environmental conditions are changed.

Essential revisions:

The authors have significantly reshaped the manuscript and added new interesting simulations in response to the reviewers' comments. We think the sensitivity analysis to different parameters, as well as the tests with alternative models, is important in showing the generality of the results.

If the authors made considerable efforts in explaining the model hypotheses, we found ourselves still puzzled about a few points in the main text, and reading the methods only provided part of the answers. We think some corrections are needed, in particular to help the reader understand how and when clustering of ameboid cells enhances chemotaxis.

1) The multiple scales at which different properties are defined makes it still difficult to figure the model out. Definition of the cell-to-cell contact energy J_{c,c} (subsection “2.1 Model setup”) and of averages in Figure 3 would help. The transition between the site and cell scale seems to be problematic if more than one cell have the same identifying string, which would happen if mutations do not happen (subsection “4.2 Evolutionary dynamics”), or if connectedness within cells is not ensured (subsection “4.1 Cell dynamics”).

2) We do not see how negative surface tension may imply 'repulsion' (subsection “Evolutionary model”, subsection “The evolution of uni- or multicellular strategies depends on environment stability”, Discussion section) between cells, rather than just an average higher probability for sites at the cell surface of sticking to the medium than to other cells. 'dispersion', also, may be due to amplification of fluctuations by persistence on the time scale of cell velocity update. Description of the behaviour of cells in isolation, especially how cell displacement depends on the magnitude of negative gamma would be very useful.

3) We do not understand the explanation for the evolution of small cell-to-cell adhesion for high frequency of environmental change. The authors claim that clusters always migrate faster up the gradient than single cells (Discussion section), but then in subsection “The evolution of uni- or multicellular strategies depends on environment stability” they seem to indicate the opposite. In the present formulation it is not clear if the advantage of single cells is given by the growing importance of the transient to clustering after reproduction and culling (that we imagine introduces, willing or not, a sort of local dispersal by creating 'holes' in coherent clusters), or by the fact that moving fast in one direction might not be the best strategy when such direction changes very fast (alike environmental response vs bet-hedging).

4) We wonder if the fact that the string defining adhesion to the medium is a substring of that defining adhesion to other cells may have evolutionary effects, as the two kinds of adhesion will be in general correlated. We understand that this is a convenient choice, but are not sure that the existence of such correlations may be justified for cells. Secondly, by looking at Appendix Figure A2.1, we are puzzled by the statement that cell–cell variation in adhesion strength after evolution is small (subsection “The evolution of uni- or multicellular strategies depends on environment stability” and Appendix subsection “2.1 Adhesion strength distribution for τs = 100 x 103 MCS”), since no quantitative comparison is made with other situations (for instance, when the unicellular strategy evolves?).

---

## [Author Response]

Summary:In this paper, the authors consider the problem of evolutionary transitions to multicellularity, and in particular the case in which aggregation drives the process. Inspired by the life cycle of *Dictyostelium*, they consider a model in which cells (moving on a grid) search for resources and can adhere to each other based on the match between ligand and receptors on their surfaces. All of this takes place in the context of a chemotactic march towards a local chemoattractant within one temporal "season", after which fitness-dependent reproduction occurs, the population is culled back to its starting size, and the environmental conditions are changed.The reviewers all are of the opinion that this work provides an interesting perspective on a possible mechanistic basis of 'collective-level' function, that stems from physical interactions among cells in the absence of explicitly modelled costs and benefits of single cell's choices. At the same time, the reviewers were clear that there are many aspects of the model and the modelling approach that are not clear, unnecessarily complicated or not well justified. In light of these, major revisions to the paper will be necessary, as explained below.

We are glad that reviewers found our work interesting – the summary nicely captures the message of the paper. We extensively revised the manuscript after the reviewers comments and suggestions.

The largest changes to the manuscript are summarised here, and below we respond to reviewers remarks point by point.

We have greatly expanded the model specification in the main text, and explain the model in much more detail in the Materials and methods section – including a clarification of parameter values. We have also run a large number of new experiments, added a robustness analysis of both collective chemotaxis and the evolutionary dynamics, we checked that our results hold when a different model of emergent chemotaxis is implemented and when costs are added. Finally, we discuss our results more in depth.

Essential revisions:1) Considering the paper as a whole, there are far too many things happening at once to draw any meaningful conclusions. There is the complexity of adhesion, the nature of the chemotaxis, the temporal switching between seasons, and the reproduction process. Each of these is explored to a limited extent, and it is unclear which are absolutely crucial to the conclusions reached and how sensitive the conclusions are to the assumptions made.

Although the model is somewhat complex in the implementation details, its key ingredients are – as the reviewers note – just four: (evolvable) adhesion, chemotaxis, environmental switching and replication. Conceptually, these are the minimal elements for a mechanistic evolutionary model: the adhesion properties are inherited with mutation upon replication, and the environment switch after reproduction allows for continual selection on reaching the peak of the gradient by chemotaxis.

We now explicitly mention these four elements in the Introduction.

We revised and expanded the model definition in the main text (subsection “2.1 Model setup”) and the Model section (see also point 2) and ran several additional simulations to show that results are robust to parameter changes and to some structural changes to the model (see Point 5 and 7 for details).

We also expanded the literature reviewed in the Discussion section and revised the Introduction to better motivate the approach we used for our model.

2) Regarding the definition of the model itself, the reviewers find it inappropriate to relegate so much of that explanation to the Materials and methods section. The very large number of parameters (18) in Table 1 needs to be made clear (and that table should be referenced – it does not appear to be at present). Please explain more of the model in the body of the paper.

We expanded the model definition in the main text and reworded it to make the principles of the model clearer. We tried to strike a better balance between giving enough details and maintaining the exposition of the model descriptive enough for a more intuitive understanding. The parameter table is now referenced in the main text (subsection “2.1 Model setup”) and in the Materials and methods section and the parameters referenced in the table are introduced in the main text.

3) The reviewers are supportive of abstract models, but inasmuch as the authors have set up a physical/biological scenario with familiar processes (chemotaxis, adhesion) it would have been very helpful to have justified the kinds of dimensionless parameters that characterize the model in terms of real physical and biological features.

The reviewers are correct that our system incorporates some biophysical elements into a simple evolutionary model. The values of some parameters, such as those for migration, are taken from published literature where the Cellular Potts Model is calibrated against empirically determined eukaryotic cell behaviour; other parameters, e.g. those of cell adhesion and volume constraint, do not have a direct correspondence to a biophysical property of the cell, but are a coarse-grained description of some cell properties. Therefore, their value cannot be directly taken from experiments, but is instead inferred from model behaviour and is in scaling relationships with other parameters. For the “temperature” parameter please see point 4). We revised the Materials and methods section in all the paragraphs where parameters are introduced to give the details of the numerical values of the parameters: we added references when these values are taken from literature, and motivate our choices for the other parameters. We also added references that explain both the connection of these macroscopic parameters to the biophysics of the cell and their scaling relationships.

4) The essence of a Monte Carlo simulation is the definition of an energy function and a temperature, which together yield a Boltzmann factor that is used to decide if an attempted step is taken. The authors do not make clear in the main body of the text that they are performing a Monte Carlo calculation (that is only specified in Subsection “4 Model”). They refer to MCS (Monte Carlo Steps) in the body of the paper without defining that term. But the larger question is why this kind of nonequilibrium biological system should have such an energy, and what would be the biological significance of the temperature? In addition, of course, the "steps" taken are those of Monte Carlo algorithm and have no direct interpretation in terms of real time.

We now explicitly state that we specifically use the Metropolis algorithm (which is a Monte Carlo method) and the rationale for the energy function in the main text (subsection 2.1 Model setup”). We also define a Monte Carlo Step (subsection 2.1 Model setup”).

With the intention of making the paper accessible to experimental researchers who might be less interested in the mathematical details, we decided to leave the formal description of the model for the Materials and methods section.

Regarding the particular choice of the energy function: it is an extension of the typical Hamiltonian used in the Cellular Potts Model, but we believe that alternative formulations would work just as well, provided that they incorporate adhesion, cell size and chemotaxis. Formulating the model in terms of its free energy makes it straightforward to extend it by adding an energy term for each biological phenomenon under study, but the basic CPM model has been shown to be mathematically equivalent to a model where overdamped Newtonian forces are specified: the energy minimization takes care of balancing the various forces acting on the cell (adhesion, volume conservation, migration, etc…) (Magno et al., 2015).

The “Boltzmann temperature” term introduces noise, and it is related to the probability that energetically unfavourable fluctuations occur. Thus, temperature scales the statistical mechanics of the model, but does not have direct biological significance.

When one is interested in finding the equilibrium state of a system, Monte Carlo Steps do not have to correspond to units of time. In our case, however, the steps of the algorithm have physical significance, because they correspond to changes caused by the forces acting on a cell. This, together with using the Metropolis algorithm and a low enough temperature, guarantees that every Monte Carlo step lasts effectively the same amount of physical time in our model, so that we can use the model for dynamical simulations (see Glazier, Balter and Poplawski, 2007, and for a broader discussion e.g. Newman and Barkema 1999).

We now clarify the energy function and its different terms more extensively in the Materials and methods section, including a better explanation of the temperature parameter.

We also added references to literature that delves deeper in the statistical mechanics (subsection “4.1 Cell dymanics”) and the biophysical motivation for the construction of the various terms of the energy function.

5) The presentation of the model and the main results lack clarity in some key aspects:a) the relation between cell–cell and cell-medium adhesion and surface tension (subsection “Strongly adhering cells perform efficient collective chemotaxis We first assessed how well groups of cells with different adhesion strengths”) is not explained, so it is not really clear what negative surface tension means.

We now explain the connection between adhesion and surface tension in the model introduction of the main text (subsection “2.1 Model setup”).

b) as surface tension pools two different kinds of adhesion, does it mean that in a certain sense adhesion to the surface can be traded off against adhesion between cells? This is important to know in connection to experiments.

Yes, in the sense that adhesion between cells can occur when cells form stronger junctions, or when they adhere less strongly to the medium.

We now mention this in the main text (subsection “2.1 Model setup”).

However, the medium in our model is not a substrate on top of which cells live. Modelling such substrate is possible (see e.g. van Oers et al., 2014), but would have further complicated the model.

c) since the measure of sequence complementarity is symmetric, why does one need to suppose the existence of both a ligand and a receptor? Would it change anything if cells were characterized by only one sequence? If yes, it would be interesting to know if at the end of the numerical experiment ligand and receptor evolve to be the same or if 'molecular' diversity is maintained.

Separate ligand and receptor are needed for versatility in cell adhesion. To see why, consider that with a one-sequence complementary binding system, two cells with identical sequence, such as two cells after division, would always strongly repel (or depending on the implementation, strongly adhere). Instead, with separate ligand and receptor, two identical cells can have any binding strength (also three cells may have any combination of adhesion strength, etc…).

This versatility might be possible also with a one-sequence system, if more complex rules were implemented. We chose these ones for the sake of simplicity, versatility and because they are a more general framework (which we are currently further developing) that allows differential adhesion between cell types.

We now make clear in the main text how the ligand-receptor system is important for versatility in adhesion (subsection “2.1 Model setup”).

Moreover, the reviewers comment prompted us to actually measure the diversity in adhesion energies when cells adhere. We now show that the distribution of adhesion energy is uni-modal, indicating that there is essentially one phenotype, and thus any molecular diversity that still allows for adhesion is selectively neutral, in agreement with the fact that we observe only one cluster at evolutionary steady state.

We now show this in Supplementary Material (Supp. Section S9) and refer to it in the main text (subsection “2.1 Model setup”).

d) the process of cell division/regrowth and the fact that cells do not change position from one season to the next should be more clearly explained in the main text.

We now clarify it in the main text (subsection “2.1 Model setup”)

e) what is the initial spatial distribution of cells at the beginning of every season, and if this matters (many models assume aggregation-dispersal cycles, that does not seem to be the case here), should be specified or repeated in the evolutionary section.

At the beginning of a new season cells retain their previous spatial distribution, i.e. we do not include dispersal. We now make this clearer in the main text both in the model introduction and in the evolutionary section (subsection “2.1 Model setup”).

f) Figure 5 should depict a case of bistability: now it is not clear that different evolutionary outcomes are associated with differences in the initial surface tension, rather than in the initial cell configuration. It would by the way be interesting to see if the second also gives rise to bistability.

Bistability depends on the initial surface tension in Figure 5 in the sense that we initialise a population of identical cells with a predetermined initial adhesion energy, and we let evolution proceed from there.

Within each season, both the spatial distribution and the surface tension of the resident population and the mutant are important, as they co-determine the direction of evolution. However, the spatial distribution of the cells in the evolutionary simulation is not a free parameter, but an outcome of the model dynamics.

This is what we aimed to show in Figure 6C and 6D.

We could, however, initialise an evolutionary experiment with a bimodal population, such as those in Figure 6A and B – which could correspond to a case where a population (rather than mutants) is invading the space of another. In that case, the initial configuration would be critical for the first season: the population that begins closer to the gradient wins. After this one season, the adhesion energy of the resident population will determine the evolutionary dynamics.

We revised the paragraph to make this clear (subsection “2.1 Model setup”), and along with this we changed the order of the figure, so that what was C,D (representing a situation more similar to the evolutionary dynamics) are now A,B and vice versa (the content of the figure is unchanged).

We agree with the reviewers that trying other initial cell configurations, such as dispersal, would be interesting: we make the case that bistability would still arise in the Discussion section.

6) Cell migration (subsection “Cell migration”) is defined in terms of the actual direction of the cell over the past steps. This seems to build in persistence, and would appear to have a profound effect on the dynamics. Is this the case?

Yes, short-term persistence is important for the collective dynamics. As cells keep their direction of motion for a short while (i.e. for their persistence time = 50 MCS), their pushing forces on each other are resolved when cells align and form streams within the cluster.

We now clarify the effect of persistence on collective behaviour in the subsection “2.1 Model setup” and Discussion section; where we also added some additional references. We also ran new simulations to check different values of persistence strength (see also remark 7).

This also ties into the improved chemotactic migration of clusters of cells and whether this is due to the law of large numbers (see reply to remark 8).

7) In general, it would be useful if statements like "In our case, aggregation leads to a highly efficient search strategy, guided by long-range, albeit noisy, gradients." (Discussion section) could be made more quantitative. For instance, one would like to get a sense of whether the conclusions are robust to changes in (at least a few important) parameters. One would expect so from results in active matter physics, but it would be useful of the authors could argument it and indicate when they expect different conclusions to hold. Moreover, what is the role of the particular gradient chosen here in 'focalizing' the formation of multicellular groups (would an essentially 1-D gradient, where isolines are parallel, do the job?) and of its intensity/spatial variation (in the movie, one sees that the centre of the gradient changes among four positions, does it matter?).

We removed the statement, and we revised the paragraph to make clearer what we meant, as we added some references (see point 12).

We also ran several simulations to assess robustness. We studied the robustness of individual vs. collective behaviour by varying chemotactic strength, persistent migration strength, cell size (which was already in the paper, but it is now clarified) and population size (for more on population size see point 9).

We find (and report in the main text – subsection “2.1 Model setup”, and Appendix 1 subsection “1.5 Robustness of collective chemotaxis to changes in persistence strength” and subsection “1.6 Robustness of collective chemotaxis to changes in chemotactic strength”) that, qualitatively, results are robust to migration parameter changes: collective chemotaxis is faster than single cell chemotaxis. Quantitatively, larger persistence strength increases the speed of collective chemotaxis more than chemotactic strength.

We then checked that the outcome of the evolutionary dynamics were also robust to changes in migration parameters. We find that it still holds that adhesion evolves when seasons are sufficiently long, and otherwise the single-cell strategy evolves.

We also ran evolutionary simulations with a 1D gradient (with parallel isoclines) and with a steeper gradient. Results change quantitatively but not qualitatively, with a larger change in the case of a steeper (more precise) gradient.

We report on this in the main text (subsection “2.1 Model setup”), and we have added Supplementary sections S10, S11 and S12, where we show the new data.

Lastly, the four positions of the peak of the gradient were chosen for visual clarity, and the evolutionary dynamics would be identical if peak location was chosen differently (because individual cells are large enough not to sense the lattice anisotropy, and thus it makes no difference where the gradient is).

8) The authors claim that, in contrast to previous work, the increased fitness of the aggregates (better ability to perform chemotaxis) is an emergent property. The reviewers struggled to find a physical/mathematical explanation as to why such a relationship exists in the model but it appears that subsection “Chemotaxis” contains the mechanism. The text speaks of the "centre of mass of the perceived gradient". Unless we are mistaken, such a quantity averages over the individual constituent's contributions in such a way that larger cells will have more accurate measurements of the gradient. This is just the law of large numbers. If this is the case, then this feature is not an emergent property at all, but is part of the definition of the model. Please clarify. If the above critique is correct, then why bother with the complex model? The authors could just use the fact that larger aggregates are better at chemotaxis for the reason given and proceed from there.The above suggests that the authors have basically put the answer in from the beginning. The model has the explicit feature that those that perform chemotaxis better reproduce more. So of course, that will be reinforced. But multicellularity has costs and benefits, and the model does not appear to contain any costs associated with multicellularity. In real biological examples there are many – the increased metabolic cost of the structures that hold cells together, greater need for regulatory genetic networks, etc.

The reviewers raise two important points, that together form the main argument of the paper: Is collective behaviour really emergent or rather built-in? And if so, does it trivially follow that collective chemotaxis drives the evolution of multicellularity? They also point out that additional costs may be paid by cells in multicellular aggregates. We respond to these three points (collective chemotaxis, evolution, costs) in order.

The reviewers are correct that with our implementation of chemotaxis, larger cells will measure the gradient more accurately; we agree that chemotaxis of individual cells is not an emergent property, and that cluster chemotaxis results from chemotaxis of individual cells. However, chemotaxis of cell aggregates does not straightforwardly follow the law of large numbers in our model; in fact, we find that clusters of many small cells perform worse than a single large cell with the same total size (we mention this in subsection “2.1 Model setup”, and show it in Suppl. Section S7), and that chemotaxis scales sublinearly with of cluster size (which we now explain in the main text – subsection “2.1 Model setup” and show in Supp. Section S2).

This is because neither our cells nor our clusters are rigid, and there is no explicit mechanism for transferring gradient information between cells. It is therefore not immediately obvious that a larger number of cells have a “shared” knowledge of the gradient in the same way that one big cell has precise knowledge of all of the gradient it covers. In our model, cells convey gradient information through emergent collective streaming (see response to point 6, and earlier work by Smeets et al., 2016, Beltman et al., 2006, Szabo et al., 2006), which becomes biased towards the weak chemotactic signal. We now mention this in the results (subsection “2.1 Model setup”), add these references and come back to it in the Discussion section.

When cells form rigid clusters instead, collective migration is tantamount to the law of large numbers – and therefore also built-in (as assumed in e.g. Varennes et al., 2017) because a cell’s polarisation towards the perceived gradient translates linearly and instantaneously to cluster movements. We thank the reviewers for pointing this out to us, as it constitutes a good reference model to compare our results to.

To completely make sure that our results do not depend on the effect of the law of large numbers, we ran a few simulations in which we implemented an alternative mechanism of collective chemotaxis (proposed in Camley et al., 2016). With this mechanism individual cells do not sense the gradient, only the local concentration of the signal. This entailed a small modification of the Hamiltonian to introduce contact inhibition of locomotion (CIL) and higher membrane polarisation rate when exposed to higher concentration of the chemoattractant. Our evolutionary results are remarkably robust to the particular implementation of collective chemotaxis. We now describe this in subsection “2.1 Model setup” and show the data in Supplementary Section S14, as well as comment on this in the Discussion section.

Regarding the evolutionary dynamics:

We agree with the reviewer that in the experiment as we have set it up, potential selection of multicellularity could be expected a priori, because clusters of cells always perform chemotaxis better in our model. We write that we do not explicitly select for multicellularity because the model rewards cells for being in a group only implicitly. This is different from many previous models where such a term is explicitly present in the equation to calculate fitness (e.g. in terms of shared benefits or in terms of decreased chance of predation).

Instead, selection acts only individual cells to increase their reproductive success: i.e. some cells will be close to the source of the gradient and reproduce more, but this does not need to happen through collective behaviour – even though adhering aggregates are always better than individual cells at chemotaxis. Even if we were able to achieve full open-ended evolution in a mathematical model, it would always be possible to argue that the result has been put in 'a priori' if selection happens to stumble upon mechanisms that we knew existed a priori in another system. It would thus philosophically be impossible to formally claim the system has found a 'novel' solution. Interestingly, in our set up, collective chemotaxis does not evolve when seasons are short, and instead the opposite strategy is selected because it leads to fast scattering, so that some cells will have higher fitness.

This shows an important distinction between fitness function (which cells replicate) and the evolutionary strategy, i.e. how cells organise through evolution as a response to the fitness function.

We agree that this is a more nuanced point that wasn’t very clear in the manuscript, so we now explain this more extensively in the Discussion section.

Lastly, regarding the lack of costs in the model: we agree with the reviewers that multicellularity may come with costs. We originally did not include them to limit model complexity. However, we took up the reviewers’ suggestion and ran a set of simulations in which we incorporated a cost for adhering to other cells. Costs are as follows: cells that spend most of the season adhering to other cells receive a fitness penalty proportional to the amount of time and membrane in contact.

Results are robust to this change: long seasons lead to multicellularity, provided that costs are not too high – with the evolutionary transition to multicellularity shifted to longer season duration with larger costs. We now present these new results in the main text (subsection “2.1 Model setup"), and give more details about implementation and results in Supplementary Section S15.

9) The referencing of the text to Figure 3 is all mixed up, leaving both text and figure hard to follow. -The authors should revise this section and make sure that they clearly state if higher chemotactic performance arises due to longer persistence of cell clusters only or due to longer persistence and higher chemotactic accuracy of whole cell cluster. Varennes et al., (2017) and manuscripts citing this work give measures for chemotactic accuracy within cell populations. – Figure 3D should show error bars. Annotation of Figure 3F should be detailed, what is bar{X}? Is this the local gradient including noise or averaged on which scale.

We revised the paragraph to improve its clarity, and the references are now in order.

We now clarify that the higher performance of cells in a cluster relative to single cells comes about from collective streaming of these cells, which increases persistence and biases migration towards the peak of the gradient (see also responses to point 6 and 8).

Moreover, we report on the robustness analysis of chemotactic parameters (see response to point 7).

Regarding accuracy measures: we now use the model proposed in Varennes et al., (2017) as a reference model, to explain why our results differ (see also point 8).

We present this in the main text (subsection “2.1 Model setup"), and show it in the Supplementary Section S2.

Regarding error bars in Figure 3D: error bars around numerically inferred derivatives are a notoriously difficult problem, as data fluctuations become over-amplified. Diffusive exponent calculations – which rely on fitting a power law to the data (and thus allows error estimates), are also often inaccurate with anomalous diffusion.

To show more transparently that the plot represents the data accurately, we clarify the intermediate steps from the MSD plot (Figure 3C) to Figure 3C, in a new Supplementary Material section S1.

We improved the annotation of the Figure 3F (X is the “true” direction of the gradient (the direction from the centre of mass of the cell to the peak of the gradient), x is the direction perceived by individual cells).

10) The assessment time scale emerges as a decisive factor – it appears as a theoretical construct right now. What could it correspond to in the real world? Please discuss.

It corresponds to the stability/volatility of resources. The idea behind it is that resources that persist in one place for sufficiently long can be traced up their chemokine gradient (this selects for multicellular behaviour because a group does chemotaxis better than individuals). When resources are volatile and the associated gradient does not persist for long, a search strategy based on spreading out allows some cells to reach them (which selects for uni-cellular strategy).

Resources are modelled only implicitly – through the chemoattractant gradient they generate and through how long they persist.

The precise seasonality of these resources might be a good assumptions if resources are deposited in the system by periodic phenomena (e.g. tides, or yearly cycles), but might be unrealistic for some types of resources, and some stochasticity will be introduced in future work. However, we think that if the distribution of resource persistence is sufficiently well behaved (e.g. if its mean exists and it has a relatively small variance compared to the rate of cell evolution), then cell evolution would converge to the average season duration.

We now clarify what the season duration corresponds to in the main text (subsection “2.1 Model setup”) and discuss its limitations in the Discussion section.

11) As for the particular details of the model, it is left unsaid in the main text but stated in the Materials and methods section that there is a preferred cell size A_T and a harmonic energy around that size. As the target size is (Table 1) some 50 pixels, we are confused, as it seems that each "cell" occupies one lattice size. This energy would then clearly bias the system to aggregate already. Please clarify. The use of the term "pixel" for a lattice site is confusing.

We now clarify that cells occupy more than one lattice site in the main text (subsection “2.1 Model setup”). The energy term proportional to (A – A_T)^2 is indeed responsible for maintaining cell size around the value A_T = 50 lattice sites, by energetically dis-favouring cell size deviations from A_T (we now clarify this in subsection “2.1 Model setup”).

We also substituted the term lattice site to the term pixel throughout the text and figures.

12) The literature overview appears limited – please revise and consider recent work for example but not limited to Varennes et al., (2017); Jacobeen et al., (2018). The authors should also discuss Guttal and Couzin, 2010. And they should acknowledge relevant literature exploring, for example, similar issues in the Volvocales; Solari er al., (2006); Solari, Kessler and Goldstein, (2013).

We thank the reviewers for the additional references. They are now integrated in the Introduction and Discussion section.

[Editors' note: further revisions were suggested prior to acceptance, as described below.]

Summary:In this paper, the authors consider the problem of evolutionary transitions to multicellularity, and in particular the case in which aggregation drives the process. Inspired by the life cycle of *Dictyostelium*, they consider a model in which cells (moving on a grid) search for resources and can adhere to each other based on the match between ligand and receptors on their surfaces. All of this takes place in the context of a chemotactic march towards a local chemoattractant within one temporal "season", after which fitness-dependent reproduction occurs, the population is culled back to its starting size, and the environmental conditions are changed.Essential revisions:The authors have significantly reshaped the manuscript and added new interesting simulations in response to the reviewers' comments. We think the sensitivity analysis to different parameters, as well as the tests with alternative models, is important in showing the generality of the results.If the authors made considerable efforts in explaining the model hypotheses, we found ourselves still puzzled about a few points in the main text, and reading the methods only provided part of the answers. We think some corrections are needed, in particular to help the reader understand how and when clustering of ameboid cells enhances chemotaxis.

We have added clarifications and more details in several parts of the paper, we have also included two new videos taken from a simulation with one cell and a simulation with a population of non-adhering cells.

In the following, we respond to each question of the reviewers in detail.

1) The multiple scales at which different properties are defined makes it still difficult to figure the model out. Definition of the cell-to-cell contact energy J_{c,c} (subsection “2.1 Model setup”) and of averages in Figure 3 would help. The transition between the site and cell scale seems to be problematic if more than one cell have the same identifying string, which would happen if mutations do not happen (subsection “4.2 Evolutionary dynamics”), or if connectedness within cells is not ensured (subsection “4.1 Cell dynamics”).

We now make clear that J_cc is the energy between two adjacent lattice sites belonging to different cells (subsection “2.1 Model setup”), and that average J is obtained by averaging the J values extracted from the simulation, after steady state is reached (see caption of Figure 5).

The ligand and receptor strings are not used as an identifier for cells, therefore two cells can have the same bitstrings (e.g. as an outcome of cell division without mutations). The identifier of a cell ‘c’ is the number ‘s’ (a positive integer) which is also used as the spin value for the lattice sites composing the cell. It is by definition different between cells. With N cells there are N+1 different spin values (N cells plus the medium, which is given spin=0). In practice, this means that for each cell, the unique identifier (and the spin value of its composing lattice sites) is a positive integer.

After cell division, the lattice sites composing two cells are given different spin values. Moreover, in the Hamiltonian, the function for adhesion energy includes a (1 – Kronecker delta) term, which calculates adhesion energy only between lattice sites that have different spins (i.e. only at the cell surface).

Regarding connectivity: the chances that a cell loses connectivity with some of its lattice sites is rare, and when it happens it is quickly resolved because disconnected pixels have large contact with the medium (or with another cell) and thus large energy.

We now state that each cell is given a unique identifier corresponding to the spin value, at the beginning of the model presentation in the main text (subsection “2.1 Model setup”).

We clarify that the spin value of lattice sites is used as the identification number of cells in the Materials and methods section.

We now also state that a new spin value is assigned after cell division to ensure uniqueness of cell identity (subsection “Replication”).

2) We do not see how negative surface tension may imply 'repulsion' (subsection “Evolutionary model”, subsection “The evolution of uni- or multicellular strategies depends on environment stability”, Discussion section) between cells, rather than just an average higher probability for sites at the cell surface of sticking to the medium than to other cells. 'dispersion', also, may be due to amplification of fluctuations by persistence on the time scale of cell velocity update. Description of the behaviour of cells in isolation, especially how cell displacement depends on the magnitude of negative gamma would be very useful.

We agree with the reviewers that “repulsion” is a confusing term, and we substitute it with “dispersion” throughout the text.

As the reviewers suggest, negative surface tension implies that cells prefer contact with the medium over other cells, which leads to dispersal. Dispersal is faster for cells that do not adhere (gamma<0) than for cells that adhere neutrally (gamma=0), because cell–cell contact is energetically costly for negative gamma. The reason is that medium must penetrate in between two cells for them to detach, which is a random, energetically neutral process for gamma=0. For both gamma<0 and gamma=0 persistent cell migration speeds up dispersion. However, we see no difference in chemotaxis between the two cases (Figure 3A), indicating that the dispersal phase is very short. Once cells are sufficiently far apart, the gamma value no longer impacts their displacement because their behaviour depends only on J_{c,m}, which is constant in the chemotactic experiments (this is always the case for simulations run with only one cell).

We now mention in the main text that cells initially in a cluster disperse for gamma=0 and gamma<0 and once in isolation each cell behaves like those from simulations run with a single cell (i.e. as those in Figure 3“one cell”) (subsection “Strongly adhering cells perform efficient collective chemotaxis”).

We also include a video with an individual cell (Video 1) and one with a group of cells that do not adhere (Video 3).

We also refer to the video of a cluster of adhering cells in the same paragraph, in order to facilitate the comparison between these cases (subsection “Strongly adhering cells perform efficient collective chemotaxis”).

3) We do not understand the explanation for the evolution of small cell-to-cell adhesion for high frequency of environmental change. The authors claim that clusters always migrate faster up the gradient than single cells (Discussion section), but then in subsection “The evolution of uni- or multicellular strategies depends on environment stability” they seem to indicate the opposite. In the present formulation it is not clear if the advantage of single cells is given by the growing importance of the transient to clustering after reproduction and culling (that we imagine introduces, willing or not, a sort of local dispersal by creating 'holes' in coherent clusters), or by the fact that moving fast in one direction might not be the best strategy when such direction changes very fast (alike environmental response vs bet-hedging).

The single-cell strategy can be considered a type of bet-hedging: over multiple seasons cells spread out throughout the field (negative gamma ensures that cells do not cluster after division). By chance (and aided by inefficient chemotaxis), some cells will be located near the peak of the gradient at the end of every season. This strategy is at advantage when seasons change rapidly because a multicellular cluster does not have the time to reach the peak of the gradient and, over multiple seasons (and over multiple switching of the gradient direction) will end up at the centre of the field.

We now discuss this in the main text (subsection “The evolution of uni- or multicellular strategies depends on season duration”) and clarify the Discussion section and include two new videos (Video 6 and Video 7) that show the different behaviour of a cluster of adhering and non-adhering cells over a couple of short season (duration = 10000 times steps). In order to make the population dynamics clearer, we set mutation rate = 0 in both videos, and all cells are initialised with the same ligands and receptors (gamma = -4 in one video and gamma = 6 in the other).

These videos also address the reviewer’s remark regarding the evolutionary consequences of culling the population after cell division. Adhering cells indeed experience this transient local dispersal (far more than non-adhering ones, as the latter are already dispersed at the beginning of each season). As a consequence, adhering cells have a short period of about 2000 time steps in which collective migration is not as efficient as for a fully connected cluster.

We therefore hypothesise that in absence of such culling, the point at which multicellularity evolves shifts to slightly shorter seasons (about 2000 time steps shorter). The culling alone therefore does not explain the evolution of the unicellular strategy.

We now include this discussion in the main text (subsection “The evolution of uni- or multicellular strategies depends on season duration”).

4) We wonder if the fact that the string defining adhesion to the medium is a substring of that defining adhesion to other cells may have evolutionary effects, as the two kinds of adhesion will be in general correlated. We understand that this is a convenient choice, but are not sure that the existence of such correlations may be justified for cells. Secondly, by looking at Appendix Figure A2.1, we are puzzled by the statement that cell–cell variation in adhesion strength after evolution is small (subsection “The evolution of uni- or multicellular strategies depends on environment stability” and Appendix subsection “2.1 Adhesion strength distribution for τs = 100 x 103 MCS”), since no quantitative comparison is made with other situations (for instance, when the unicellular strategy evolves?).

We agree with the reviewers that using the same string for cell–cell and cell-medium adhesion could potentially have odd evolutionary consequences. We reasoned that if such correlations played an important role, the value of gamma from randomly generated ligands and receptors should differ from zero. We tested this by generating 100000 random pairs of ligands and receptors, and found that the average value of gamma is precisely zero, and the distribution of gamma values is symmetric around it.

The reason for this effective independence between cell–cell and cell-medium adhesion is that the number of possible configurations for each combination of the observed J values is very large, much larger than the number of combinations for which the overlap between cell–cell and cell-medium bits is important.

The same dataset of randomly generated ligands and receptors also provides a good “null model” for whether the variability of adhesion strength is narrowed by the evolutionary process. We observe that the variance of adhesion strength is smaller than that observed for random pairs of ligand-receptor strings (variance is respectively about 11 vs. 3.5).

We now mention this in Appendix 2.1 and Appendix—Figure A2.1 to include the data on random ligands and receptors. To facilitate the quantitative comparison between the two distributions we also mention the numerical values of the adhesion strength mean and variance for all the different cases shown in the figure (Appendix subsection “2.1 Adhesion strength distribution for τs = 100 x 103 MCS”).